# Hyperactivation of HER2-SHCBP1-PLK1 axis promotes tumor cell mitosis and impairs trastuzumab sensitivity to gastric cancer

Wengui Shi [1,2,5], Gengyuan Zhang [3,5], Zhijian Ma[4,5], Lianshun Li[4,5], Miaomiao Liu[4], Long Qin[1,2], Zeyuan Yu[3], Lei Zhao[1,2], Yang Liu[1,2], Xue Zhang[4], Junjie Qin[1,2], Huili Ye[1,2], Xiangyan Jiang[4], Huinian Zhou[3], Hui Sun[1,2✉] & Zuoyi Jiao [1,2,3,4✉]

Trastuzumab is the backbone of HER2-directed gastric cancer therapy, but poor patient response due to insufficient cell sensitivity and drug resistance remains a clinical challenge. Here, we report that HER2 is involved in cell mitotic promotion for tumorigenesis by hyperactivating a crucial HER2-SHCBP1-PLK1 axis that drives trastuzumab sensitivity and is targeted therapeutically. SHCBP1 is an Shc1-binding protein but is detached from scaffold protein Shc1 following HER2 activation. Released SHCBP1 responds to HER2 cascade by translocating into the nucleus following Ser273 phosphorylation, and then contributing to cell mitosis regulation through binding with PLK1 to promote the phosphorylation of the mitotic interactor MISP. Meanwhile, Shc1 is recruited to HER2 for MAPK or PI3K pathways activation. Also, clinical evidence shows that increased SHCBP1 prognosticates a poor response of patients to trastuzumab therapy. Theaflavine-3, 3'-digallate (TFBG) is identified as an inhibitor of the SHCBP1-PLK1 interaction, which is a potential trastuzumab sensitizing agent and, in combination with trastuzumab, is highly efficacious in suppressing HER2-positive gastric cancer growth. These findings suggest an aberrant mitotic HER2-SHCBP1-PLK1 axis underlies trastuzumab sensitivity and offer a new strategy to combat gastric cancer.

[1] Cuiying Biomedical Research Center, Lanzhou University Second Hospital, Lanzhou, People's Republic of China. [2] Biobank of Tumors from Plateau of Gansu Province, Lanzhou University Second Hospital, Lanzhou, People's Republic of China. [3] The Department of General Surgery, Lanzhou University Second Hospital, Lanzhou, People's Republic of China. [4] The Second Clinical Medical College, Lanzhou University, Lanzhou, People's Republic of China. [5] These authors contributed equally: Wengui Shi, Gengyuan Zhang, Zhijian Ma, Lianshun Li. ✉email: yaohy04@126.com; jiaozy@lzu.edu.cn

Human epidermal growth factor receptor 2 (HER2) is often amplified or overexpressed in gastric carcinoma[1]. Through dimerization with other HER members, HER2 activates downstream pathways, including the mitogen-activated protein kinase (MAPK) and phosphatidylinositol 3-kinase (PI3K) pathways, and promotes tumorigenesis by increasing cell proliferation, metastasis, and invasion[2]. A lot of HER2-directed therapies have been used in the treatment of HER2-positive cancers. Trastuzumab, pertuzumab, lapatinib, and T-MD1 are used in HER2-positive breast cancer[3]. Trastuzumab and trastuzumab deruxtecan are effective anti-HER2 therapy showing survival benefit in gastric cancer[4–6]. In the trastuzumab for gastric cancer (ToGA) trial, the overall survival (OS) of patients with HER2-positive advanced gastric cancer was improved 2.7 months when trastuzumab was combined with conventional chemotherapy, compared to chemotherapy alone. These results led to the approval trastuzumab, which is now the first-line treatment in combination with oxaliplatin or fluorouracil chemotherapy for patients with HER2-positive metastatic gastric cancer[4]. However, many patients with HER2-positive gastric cancer still succumb to their disease following trastuzumab therapy, one of the main reasons is the intrinsic and secondary resistance.

The mechanisms underlying insufficient sensitivity of HER2-directed therapy are proposed to be aberrant activation of HER2 and downstream signaling, including amplification, upregulation, or mutation of HER2, KRAS, PIK3CA, AKT, and PTEN, which make it difficult to inhibit the activation of downstream signaling and cell growth, using trastuzumab only[7,8]. HER2-positive gastric cancer has been found to share some of these mechanisms, but also manifests specific mechanisms of resistance to trastuzumab. For example, intratumoral HER2 heterogeneity is more frequent in gastric cancer than in breast cancer, with values ranging from 23 to 79% (ref. [9]). Besides, loss of HER2 protein expression in pretreatment and posttreatment of gastric cancer patients is also a main reason of trastuzumab resistance[10]. Other proposed resistance mechanisms include alterations in HER2 downstream signaling and bypass pathways, such as upregulation of kallikrein 10 (KLK10), metastasis associated in colon cancer 1 (MACC1), C-Maf-inducing protein (CMIP), hyperactivation of HER4–YAP1 axis, TNF α-induced mucin 4 (MUC4) overexpression[11–15]. To overcome drug resistance and sensitize cells to trastuzumab, new therapeutic agents or combination therapies have recently emerged, such as afatinib (HER1, HER2, and HER3 inhibitor), trastuzumab deruxtecan (topoisomerase I inhibitor), and PI3K or MAPK pathway inhibitors[16–19]. Although trastuzumab deruxtecan has been proved resulting a survival benefit in patients with HER2-positive gastric cancer, no other new data have been obtained to date, and none of the new anti-HER2 treatment strategies improved survival significantly enough to justify registration[6,20]. Thus, identifying undiscovered molecular mechanisms underlying HER2-promoted tumorigenesis and developing new drugs to arise trastuzumab sensitivity are critical.

HER2 and other epidermal growth factor receptors (ERBBs) always employ Shc1, an important intracellular scaffold protein, to recruit cytoplasmic targets and amplify downstream signals to activate the MAPK and PI3K/AKT pathways. Importantly, Shc1 is a hub that binds with multiple interactors dynamically to direct the temporal flow of signaling information, following growth factor stimulation[21]. Here, we report that, in addition to the canonical MAPK and PI3K downstream signaling pathways, HER2 promotes tumorigenesis by direct regulation of mitotic progression through a crucial Shc1–SHCBP1–PLK1–MISP axis, which drives the sensitivity of HER2-positive cells to trastuzumab. Theaflavine-3, 3′-digallate (TFBG) can selectively inhibit SHCBP1–PLK1 complex and render gastric cancer sensitive to trastuzumab. Targeted TFBG treatment combined with trastuzumab exhibits substantial growth inhibition and tumor regression, indicating potential clinical applications in HER2-positive gastric cancer therapy.

## Results

### SHCBP1, a Shc1-binding protein, is the downstream effector of HER2.

To gain insight into the subset of Shc1-binding proteins and identify the undiscovered downstream axis of HER2, which is involved in gastric tumorigenesis, we mapped the binding proteins associated with Shc1 and screened for the interactors that were correlated with HER2 overexpression and were potential upregulated oncogenes in gastric cancer.

First, we identified the HER2-positive gastric cancer cell lines NCI-N87 and SNU-216 using western blot, immunofluorescence (IF), immunohistochemistry (IHC), and fluorescence in situ hybridization (Supplementary Fig. 1a, b). Then, we engineered an SNU-216 cell line to stably express Flag-tagged Shc1 and immunoprecipitated Flag-Shc1, following epidermal growth factor (EGF) stimulation. Using liquid chromatography–tandem mass spectrometry (LC–MS/MS) analysis, we identified 32 Shc1-binding partners, including EGFR, HER2, HER3, JUP, and SHCBP1 (Fig. 1a). To screen out if any of the binding proteins are potential HER2 downstream regulators, we carried out a gene expression correlation analysis between the identified Shc1 interactors and HER2 expression in 659 gastric cancer specimens obtained from the Gene Expression Omnibus (GEO) database, and 24 HER2-correlated Shc1-binding proteins with spearman coefficient ≥0.3 were screened out (Fig. 1b). After that, we determined if any of the identified binding proteins were potential upregulated oncogenes involved in gastric tumorigenesis. A gene expression profile of gastric cancerous and adjacent normal samples from 16 patients were performed, using mRNA microarray. We screened out five overlapping Shc1-binding proteins (JUP, EPHA2, RASAL2, LYN, and SHCBP1), which were positively correlated with HER2 expression and were upregulated in gastric cancer (Fig. 1c, d). Finally, to confirm our screening results, the mRNA expression of the identified Shc1-binding proteins in HER2-positive gastric cancer patients were detected using real-time PCR (RT-PCR). We found that SHCBP1 and RASAL2 were really upregulated in HER2-positive gastric cancer (Fig. 1e). Of the two binding proteins, we focused on SHCBP1 as it was previously reported interacting with Shc1 prior to EGF stimulation and the role of SHCBP1 in HER2-mediated signal activation was elusive[21].

To determine the screening results for SHCBP1 binding with Shc1, we transduced the cells expressing Flag-Shc1 with an HA-tagged SHCBP1 vector, and their interactions following EGF stimulation were detected, using co-immunoprecipitation and immunoblotting. We found that SHCBP1 was associated with Shc1 prior to stimulation, but was displaced following EGF treatment (Fig. 1f). In addition, using fluorescence resonance energy transfer (FRET) and immunofluorescent colocalization analysis, we showed that EGF-induced Shc1 disassociating from SHCBP1 and binding to HER2 (Fig. 1g, h and Supplementary Fig. 1c). To examine whether EGF-induced SHCBP1 and Shc1 dissociation is a cascade of HER2 activation, we pretreated cells with trastuzumab to block HER2 activation and observed that the displacement between Shc1 and SHCBP1 in response to EGF stimulation was abolished (Fig. 1i). These results were further validated by FRET analysis, demonstrating that the Shc1–SHCBP1 interaction decreased with exposure to EGF and that these effects were relieved by trastuzumab treatment (Fig. 1j).

In conclusion, these results suggest that SHCBP1 is a downstream effector of HER2 and may contribute to the

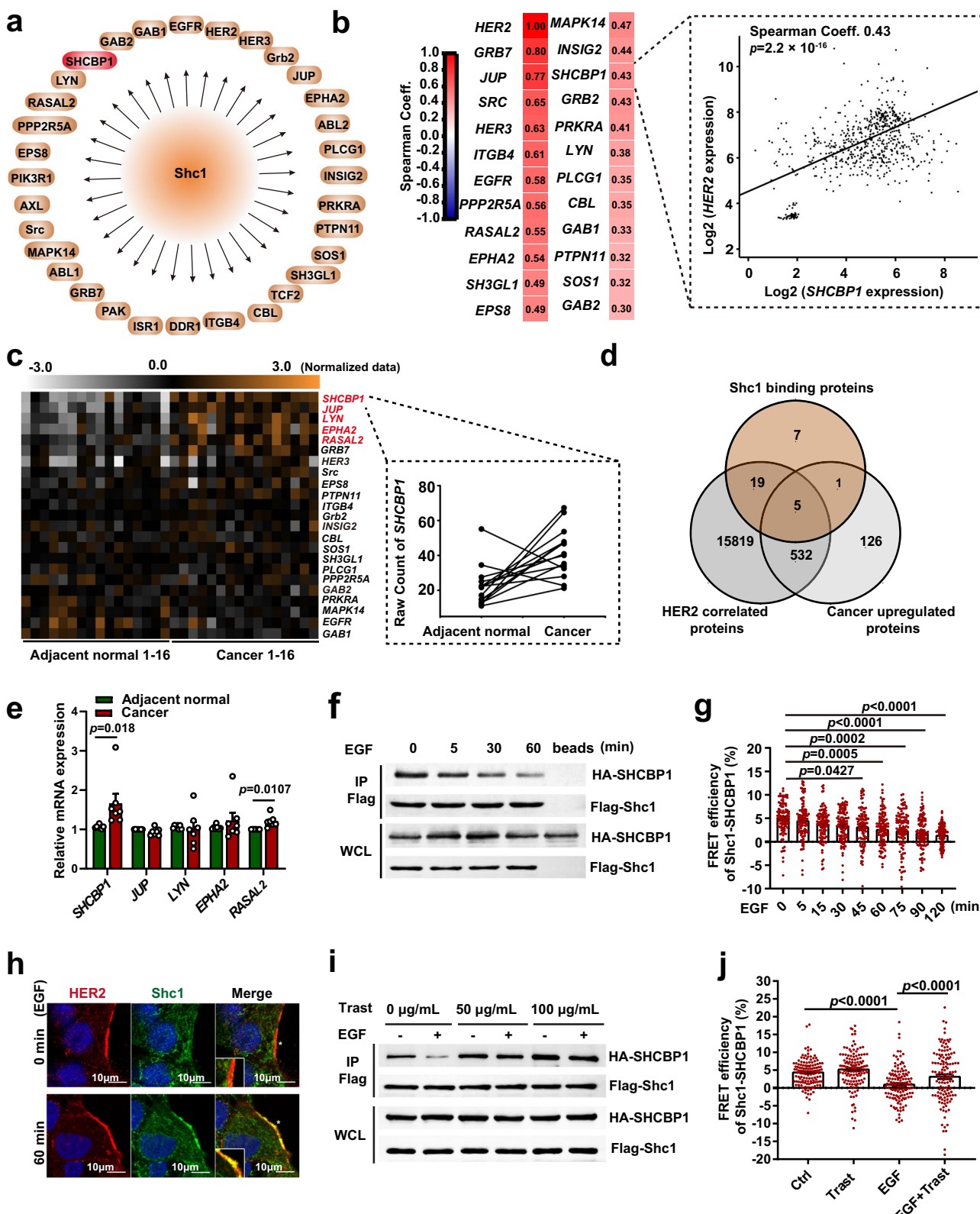

regulation of a novel oncogenic signaling axis in response to HER2 activation in gastric cancer.

**SHCBP1 is upregulated in human gastric cancer and correlates with drug sensitivity in patients subjected to trastuzumab-based therapy.** To verify whether SHCBP1 is upregulated in gastric cancer and has clinical relevance with HER2 amplification,

we performed IHC analysis of a tissue microarrays (TMAs), including 223 paired gastric cancerous and adjacent normal tissues, the results indicated that SHCBP1 was indeed highly expressed in gastric cancer samples (Fig. 2a, b). This was also confirmed by SHCBP1 immunoblotting of tissues from eight gastric cancer patients (Fig. 2c) and *SHCBP1* expression analysis of a publicly available GEO dataset (Fig. 2d). We performed

**Fig. 1 Identification of SHCBP1 as a downstream effector of HER2. a** Shc1-binding proteins in SNU-216 cells identified using liquid chromatography–tandem mass spectrometry (LC–MS/MS) analysis. **b** Gene expression correlation of Shc1-binding proteins and HER2 in 659 gastric cancer tissues from GEO data (GSE62254, GSE15459, GSE34942, and GSE54129). Representative scatter plot of *SHCBP1* and *HER2* is shown. The *p* values were determined by two-sided Spearman's rank correlation test (*n* = 659 independent biological samples). **c** Gene expression profiles of the cancer and corresponding adjacent normal tissues from 16 patients detected using mRNA microarrays. The upregulated Shc1-binding proteins are marked in red and the representative raw count of *SHCBP1* is shown. **d** A Venn diagram showing the overlap of Shc1-binding proteins, cancer upregulated proteins, and proteins correlated with HER2 expression. **e** mRNA expression of identified Shc1-binding proteins (*SHCBP1*, *JUP*, *LYN*, *EPHA2*, and *RASAL2*) in seven patients with HER2-positive gastric cancer detected by real-time PCR. Data are the mean ± standard error of the mean (s.e.m.). The *p* values were determined by paired two-sided Student's *t* test or nonparametric test (*n* = 7 independent biological samples). **f** Co-immunoprecipitation assays of Flag-Shc1 together with HA-SHCBP1 in SNU-216 cells treated with 100 ng/mL epidermal growth factor (EGF) for the indicated times. IP immunoprecipitation, WCL whole cell lysates. **g** Fluorescence resonance energy transfer (FRET) assay of eCFP-Shc1 and eYFP-SHCBP1 in cells treated with 100 ng/mL EGF for the indicated times. Data are the mean ± s.e.m. The *p* values were determined by repeated measures one-way ANOVA (*n* = 105 independent cells per group). **h** Immunofluorescence colocalization of Shc1 and HER2 in SNU-216 cells treated with/without 100 ng/mL EGF. Cells were immunostained with anti-ERBB2 antibody (red), anti-Shc1 antibody (green), and DAPI (blue). **i** Co-immunoprecipitation assays of Flag-Shc1 together with HA-SHCBP1 treated with EGF and/or trastuzumab (Trast). **j** FRET assays of eCFP-Shc1 and eYFP-SHCBP1 in cells treated with EGF and/or trastuzumab. Data are the mean ± s.e.m. The *p* values were determined by two-sided nonparametric test (*n* = 130 independent cells per group). Data of **f** and **i** are representative of at least two independent experiments. Data of **h** are representative of at least three independent experiments.

immunohistochemical and immunofluorescent analysis of the gastric cancer TMAs to determine the clinical relevance of SHCBP1 and HER2, demonstrating a weak to moderate correlation between HER2 and SHCBP1 expression (Spearman coefficient = 0.36, Fig. 2e, f). To further characterize the clinical importance of SHCBP1, we evaluated patient OS from the gastric cancer TMAs and public databases, respectively. Interestingly, high expression of SHCBP1 (*H*-score ≥ 70) was positively associated with worse OS in HER2-positive patients than in HER2-negative patients (Fig. 2g and Supplementary Fig. 2a). In addition, we also assessed the relationship between SHCBP1 and the clinicopathological characteristics of HER2-positive patients from the gastric cancer TMAs, and found a significant correlation between SHCBP1 and tumor invasion, lymph node status and tumor stage (Supplementary Fig. 2b). Furthermore, the univariate and multivariate analyses of patients from the gastric cancer TMAs demonstrated SHCBP1 expression was an independent prognostic factor for HER2-positive gastric cancer patients (Supplementary Table 1).

To investigate whether upregulated SHCBP1 confers trastuzumab sensitivity, we examined SHCBP1 status by IHC in 22 HER2-positive gastric cancer patients who received trastuzumab-based therapy (Supplementary Table 2). For these cases, patients without the evidence of disease for over 2 years after treatment with trastuzumab were considered "sensitive" and patients with death related to disease recurrence were deemed "resistant". Compared with the "sensitive" group, SHCBP1 is significantly higher in the "resistant" group (Fig. 2h, i). Furthermore, we evaluated the prognostic value of SHCBP1 in OS, showing that high SHCBP1 expression (*H*-score ≥ 70) correlates with a shorter OS time in cancer patients who received trastuzumab-based therapy (Fig. 2j).

Collectively, these findings demonstrate that the initial SHCBP1 expression significantly correlates with trastuzumab sensitivity, unraveling the clinical importance of SHCBP1 in HER2-targeted therapy for gastric cancer.

**SHCBP1 contributes to trastuzumab sensitivity in HER2-positive gastric cancer.** Given these findings, we hypothesized that SHCBP1 is a critical driver of HER2-mediated cell proliferation and is associated with gastric cancer sensitivity to trastuzumab. To explore this, we knocked down SHCBP1 expression by two different shRNAs targeting SHCBP1 in SNU-216 and NCI-N87 cells (Supplementary Fig. 3a). SHCBP1 depletion significantly suppressed cell proliferation in both NCI-N87 and SNU-216 cells (Fig. 3a, b). We also detected the dose-

dependent growth inhibition of shCtrl and SHCBP1 depletion cells in response to trastuzumab, which revealed that trastuzumab treatment caused a dose-dependent decrease in cell proliferation and knockdown of SHCBP1 significantly sensitized cells to trastuzumab both in NCI-N87 and SNU-216 cells (Fig. 3c, d). These results were further confirmed by a long-term colony formation assay (Fig. 3e). We also engineered cell lines to overexpress SHCBP1, and performed proliferation and colony formation assays, and these studies indicated that SHCBP1 overexpression reduced the sensitivity to trastuzumab in both NCI-N87 and SNU-216 cells (Fig. 3f–h).

For in vivo investigation, we established xenografts of NCI-N87 cells in nude mice and tested the efficacy of trastuzumab against tumor xenografts. As predicted by the in vitro studies, we found that trastuzumab treatment moderately slowed tumor growth with shCtrl cells, while significantly reduced tumor growth in SHCBP1-depleted cell xenograft models (Fig. 3i and Supplementary Fig. 3c, d). Consistently, IHC staining analysis of Ki-67 showed that SHCBP1 knockdown significantly enhanced trastuzumab suppression of cellular proliferation (Fig. 3j), underscoring that SHCBP1 depletion renders HER2-positive gastric cancer sensitive to trastuzumab both in vitro and in vivo. Trastuzumab combined with an SHCBP1 inhibitor may be an effective therapeutic strategy for HER2-positive gastric cancer.

**Nuclear localization of SHCBP1 contributes to HER2-mediated cell proliferation.** To explore the precise roles of SHCBP1 in trastuzumab sensitivity regulation since it departed from Shc1 following HER2 activation, we detected the subcellular localization of SHCBP1 throughout the cell division cycle in synchronized cells. Interestingly, SHCBP1 was dynamically localized to various subcellular structures during the successive steps of mitotic division. Anti-SHCBP1 antibodies produced weak, mostly cytoplasmic staining of interphase cells, but strong staining of cells in S phase and the mitotic phase. It was diffusely localized to the nuclear region during S phase and to the spindle in prometaphase and metaphase. Progressively, SHCBP1 accumulated at the central spindle during late anaphase and finally at the midbody during cytokinesis (Fig. 4a). These results suggest that SHCBP1 may translocate into the nucleus and then act as an essential mitotic component to regulate cell division. Subsequently, we assessed whether SHCBP1 nuclear localization is a signaling cascade following HER2 activation. IF staining and nucleoprotein immunoblotting analysis demonstrated that EGF induced dramatic nuclear localization of SHCBP1 in both NCI-N87 and SNU-216 cells, compared with EGF-free cells (Fig. 4b–d

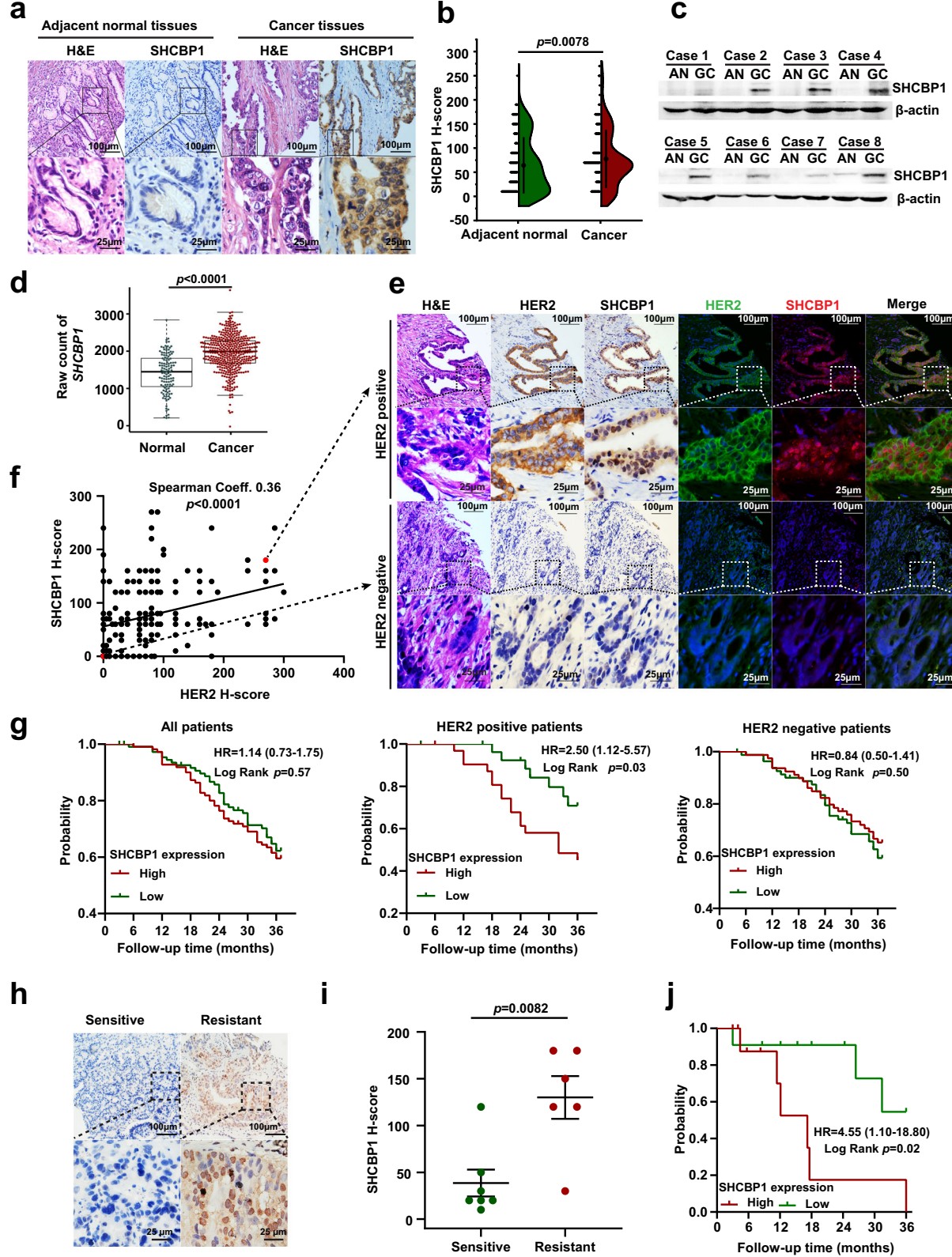

and Supplementary Fig. 4a, b). Moreover, blocking HER2 activation using trastuzumab treatment effectively abolished EGF-induced nuclear translocation (Fig. 4e, f and Supplementary Fig. 4c). These data suggest that SHCBP1 nuclear localization is a downstream consequence of HER2 activation.

To characterize the potential mechanism underlying SHCBP1 nuclear localization, we engineered SNU-216 cells stably expressing

Flag-tagged SHCBP1 deletion mutants to identify the core fragments attributed to SHCBP1 nuclear translocation. IF and immunoblotting analyses showed that the nuclear localization of the 291–562 aa fragment was significantly weaker than that of the 64–562 aa fragment, indicating that the fragment containing amino acids 64–291 was required for nuclear localization (Fig. 4g). To investigate whether a specific site modification on 64–291 aa

**Fig. 2 SHCBP1 is correlated with trastuzumab sensitivity and poor clinical outcome in human gastric cancer. a** Representative hematoxylin–eosin (H&E) and SHCBP1 immunohistochemical (IHC) staining in gastric cancer and corresponding adjacent normal tissues. **b** Quantification of SHCBP1 IHC analysis of human gastric tissue microarrays (TMAs) from 223 patients, H-score histoscore. The p values were determined by two-sided nonparametric test (n = 223 independent biological samples). **c** SHCBP1 immunoblotting analysis of gastric cancer (GC) and corresponding adjacent normal (AN) samples from eight patients. **d** Quantification of SHCBP1 expression in gastric cancer and normal specimens obtained from the GEO database (GSE66229 and GSE54129). The p values were determined by two-sided nonparametric test (n = 411 in cancer group and 121 in normal group, independent biological samples). The box plots denote medians (center lines), 25th and 75th percentiles (bounds of boxes), and minimum and maximum (whiskers). **e** Representative H&E, IHC, and immunofluorescence staining of SHCBP1 and HER2 in gastric cancer TMAs. **f** Scatter plots of HER2 versus SHCBP1 H-score in the human gastric TMAs. The p values were determined by two-sided Spearman's rank correlation test (n = 223 independent biological samples). **g** Kaplan–Meier plot of the correlation between SHCBP1 expression and patient overall survival using the human gastric cancer TMAs. The H-score < 70 is divided as SHCBP1 low expression group and H-score ≥ 70 as SHCBP1 high expression group. HR hazard ratio. The p values were determined by log-rank test (n = 223 independent biological samples). **h, i** SHCBP1 IHC staining and the quantification results in gastric cancer from patients subjected to trastuzumab-based therapy. Data are the mean ± s.e.m. The p values were determined by two-sided nonparametric test (n = 7 in sensitive group and six in resistant group, independent biological samples). **j** Kaplan–Meier plot of the correlation between SHCBP1 expression and overall survival of patients subjected to trastuzumab-based therapy. The H-score < 70 is divided as SHCBP1 low expression group and H-score ≥ 70 as SHCBP1 high expression group. The p value was determined by log-rank test (n = 22 independent biological samples). Data of **a**, **e** and **h** are representative of at least three independent experiments. Data of **c** are representative of two independent experiments.

serves for SHCBP1 nuclear translocation, we purified Flag-tagged SHCBP1 using co-immunoprecipitation following EGF treatment and analyzed the site modification by LC–MS/MS. A prominent phosphorylation site on serine 273 of SHCBP1 was identified (Fig. 4h) and further validated by mutagenesis analysis. We constructed a mutant SHCBP1 (SHCBP1$^{S273A}$), and SHCBP1 phosphorylation was detected using immunoblotting with an anti-phosphoserine (pSer) antibody. Mutagenesis analysis confirmed that EGF induced serine phosphorylation on SHCBP1$^{WT}$, but to a lesser extent on the SHCBP1$^{S273A}$ mutant (Fig. 4i). Similarly, IF and immunoblotting analyses indicated that the SHCBP1$^{S273A}$ mutant effectively blocked SHCBP1 nuclear translocation, but had no effect on EGF-induced SHCBP1–Shc1 dissociation (Fig. 4j and Supplementary Fig. 4d). Moreover, we also mutated the Ser273 of SHCBP1 to Asp273, which confirmed that SHCBP1$^{S273D}$ mutant significantly inhibited SHCBP1 nuclear translocation (Supplementary Fig. 4d, e). These results suggested that phosphorylation at the S273 site was indispensable for SHCBP1 nuclear localization.

Then, we validated the necessity of SHCBP1 nuclear localization in HER2-mediated cell proliferation. We reexpressed SHCBP1$^{WT}$ and SHCBP1$^{S273A}$ in SHCBP1 knockdown cells and detected the trastuzumab half-maximal inhibitory concentration (IC$_{50}$) in the cells. Reexpression of SHCBP1$^{WT}$ reversed the sensitization effect of SHCBP1 knockdown on trastuzumab (IC$_{50}$ = 84.04 µg/mL), while SHCBP1$^{S273A}$ reexpression did not (IC$_{50}$ = 22.47 µg/mL, Fig. 4k). Collectively, these data support our hypothesis that SHCBP1 nuclear localization is a downstream consequence of HER2 activation and contributes to HER2-mediated cell proliferation.

**HER2 mediates cell mitotic progression by activating SHCBP1–PLK1–MISP signaling pathway.** After SHCBP1 nuclear translocation, we speculated that SHCBP1 may be a mitotic protein involved in cell mitotic progression. We colocalized SHCBP1 with the centrosome and spindle, the results showed that SHCBP1 localized to the spindle poles in metaphase and to the midbody in cytokinesis (Supplementary Fig. 5a). These results suggest that SHCBP1 is a mitotic regulator, and plays pleiotropic roles in both metaphase and cytokinesis. Previous studies demonstrated that SHCBP1 regulates cytokinesis completion by the interaction with centralspindlin complex composed of MKLP1 and MgcRacGAP[22,23], but the role of SHCBP1 in metaphase is still unknown. To further investigate the mechanisms of how SHCBP1 contributes to cell division in metaphase, we knocked down the SHCBP1 expression in SNU-216 cells and detected the changes of cell mitotic progression. Cell cycle assay

using flow cytometry showed that SHCBP1 depletion induced a significant G2/M arrest (Supplementary Fig. 5b, c). We monitored the division of SHCBP1 knockdown cells by time-lapse microscopy and observed an obvious delay in mitotic progression (Supplementary Fig. 5d). Furthermore, close inspection of the mitotic spindle stained with anti-α-tubulin and the centrosomal marker γ-tubulin showed a marked increase of defective spindle and multipolar spindle in SHCBP1 knockdown cells (Supplementary Fig. 5e, f). These results suggested that SHCBP1 is essential for proper spindle formation in mitotic progression during metaphase.

Then, to investigate the detailed mechanisms of how SHCBP1 contributes to cell division in metaphase, we synchronized cells stably expressing Flag-tagged SHCBP1 to the mitotic phase using nocodazole (NOC), and identified SHCBP1-interacting proteins using immunoprecipitation and LC–MS/MS analysis (Supplementary Fig. 6a). We readily detected two mitotic proteins, PLK1 (polo-like kinase 1) and MISP (mitotic interactor and substrate of PLK1) as prominent SHCBP1-interacting proteins, and the interactions were confirmed by co-immunoprecipitation performed on NOC-arrested SNU-216 cells (Supplementary Fig. 6b–d). Interestingly, MISP has been reported as a PLK1 substrate required for proper spindle orientation and mitotic progression[24]. These findings directed our focus to the formation of SHCBP1–PLK1–MISP complex. We co-expressed Flag or HA-tagged SHCBP1, PLK1, and/or MISP in HEK293T cells, and co-immunoprecipitation assays showed that SHCBP1–PLK1–MISP complex formation could readily be observed in M phase (Fig. 5a). In a yeast two-hybrid assay, SHCBP1 interacted directly with PLK1 but not MISP, while PLK1 interacted with both SHCBP1 and MISP (Fig. 5b). The interactions were also supported by a GST pulldown assay, demonstrating that PLK1 was an intermediate protein connecting SHCBP1 and MISP for the protein complex (Fig. 5c). Subcellular colocalization of the SHCBP1–PLK1–MISP complex revealed that the complex was specifically enriched at spindle poles in metaphase (Fig. 5d). Both SHCBP1 and PLK1 accumulated at the midbody in the cytokinesis phase, but PLK1 was located at the ends, while SHCBP1 was located at the center, suggesting that the SHCBP1–PLK1–MISP complex contributes to cell division in metaphase instead of the cytokinesis phase.

Based on above findings, we reasoned that SHCBP1 may interact with PLK1 to promote PLK1 kinase activity and enhance MISP phosphorylation for cell division. To confirm that MISP is a substrate of PLK1, we synchronized cells to metaphase using NOC and observed the presence of a slower migrating MISP band in western blot, which was attributed to its phosphorylation as it

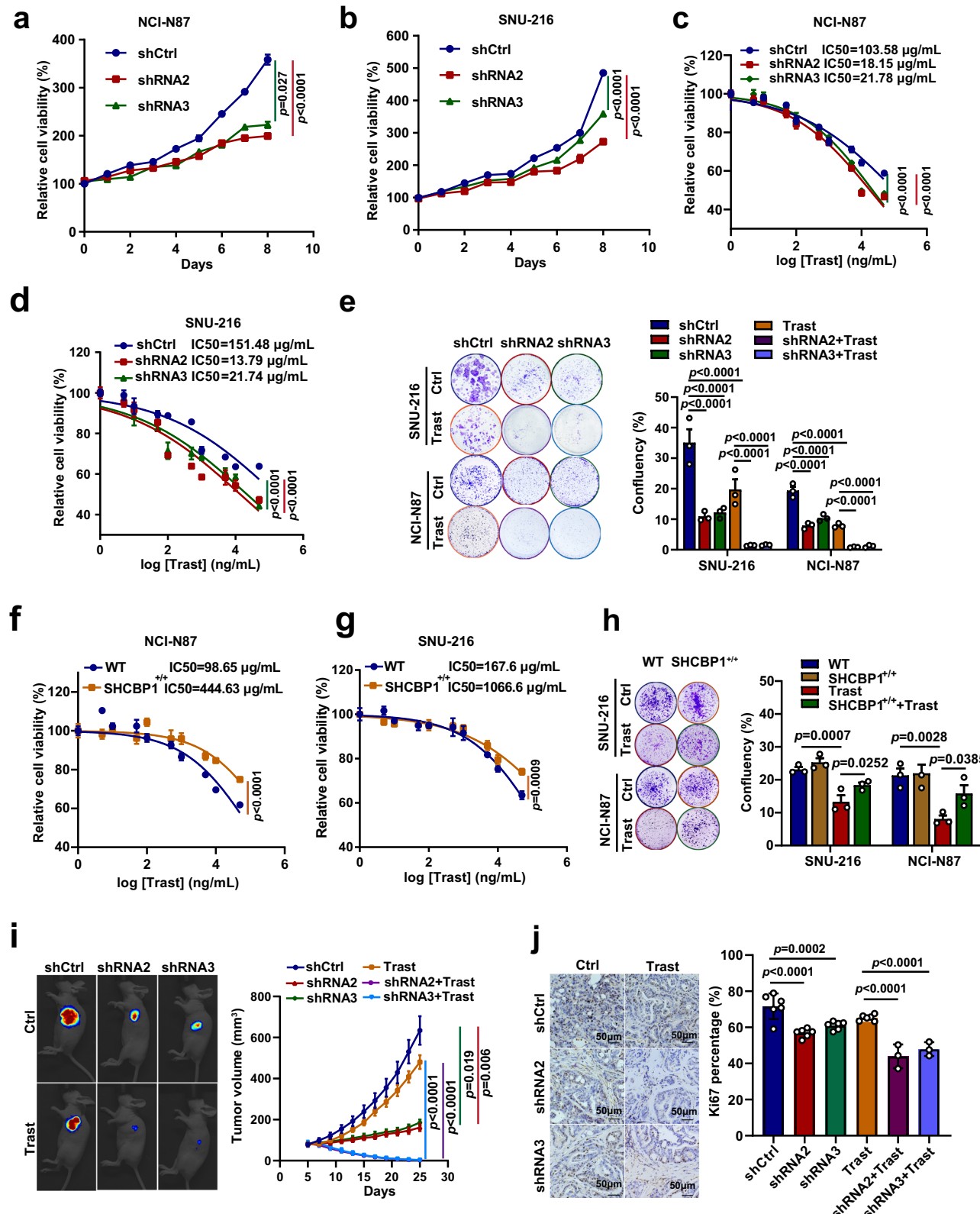

disappeared in response to calf intestinal phosphatase treatment. Overexpressed PLK1 led to a pronounced increase in the slower migrating MISP band, suggesting that PLK1 catalyzed MISP phosphorylation (Fig. 5e). Moreover, in vitro kinase assays with recombinant MISP and PLK1 were performed, and strong phosphorylation of MISP by PLK1 was detected (Fig. 5f). In order to better understand the role of SHCBP1 in the process of

PLK1 phosphorylating MISP, we detected the interaction of MISP and PLK1 in SHCBP1 knockdown cells, indicating that SHCBP1 depletion reduced the binding of PLK1 and MISP (Fig. 5g). Consistently, PLK1-induced MISP phosphorylation was also diminished in SHCBP1 knockdown cells, but reexpression of SHCBP1 reversed this inhibitory effect (Fig. 5h, i), suggesting that the activity of PLK1 toward MISP is greatly facilitated by SHCBP1.

**Fig. 3 SHCBP1 contributes to trastuzumab sensitivity in HER2-positive gastric cancer. a, b** Growth curves of NCI-N87 and SNU-216 shCtrl cells or cells with SHCBP1 knockdown. Data are the mean ± s.e.m. The *p* values were determined by two-sided nonparametric test and one-way ANOVA (*n* = 6 independent biological samples). **c, d** Sensitivity to trastuzumab (Trast) in NCI-N87 and SNU-216 Ctrl cells or SHCBP1 knockdown cells. Cells was treated with trastuzumab at the indicated concentrations for 6 days and subjected to a cell viability assay. Data are the mean ± s.e.m. The *p* values were determined by one-way ANOVA (*n* = 6 independent biological samples). **e** Colony formation and the statistical results of shCtrl cells and SHCBP1 knockdown cells in the continuous presence or absence of trastuzumab; Data are the mean ± s.e.m. The *p* values were determined by one-way ANOVA (*n* = 3 independent biological samples). **f, g** Sensitivity to trastuzumab in wild type (WT) or SHCBP1-overexpressed (SHCBP1$^{+/+}$) NCI-N87 and SNU-216 cells. Cells were treated with trastuzumab at the indicated concentrations for 6 days and subjected to a cell viability assay; data are the mean ± s.e.m. The *p* values were determined by two-sided Student's *t* test (*n* = 6 independent biological samples). **h** Colony formation and the statistical results of WT cells and SHCBP1-overexpressed cells in the continuous presence or absence of trastuzumab; data are the mean ± s.e.m. The *p* values were determined by one-way ANOVA (*n* = 3 independent biological samples). **i** Representative images (left) and tumor growth curve (right) of trastuzumab-treated xenograft mice carrying shCtrl or SHCBP1 knockdown NCI-N87 cell xenografts. Mice were administered with trastuzumab (intraperitoneal, 10 mg/kg, 2×/wk × 3) when tumors reached 100 mm$^3$ in size; data are the mean ± s.e.m. The *p* values were determined by two-sided nonparametric test (*n* = 8 independent mice per group). **j** Representative images (left) of Ki-67 immunohistochemical analysis and quantification (right) of Ki-67-positive cells of tumors from (**i**); data are the mean ± s.e.m. The *p* values were determined by one-way ANOVA (*n* = 3–6 independent biological samples).

We further investigated whether SHCBP1–PLK1 facilitated MISP phosphorylation is a downstream consequence of HER2 activation and found that EGF exposure gradually induced MISP phosphorylation, and these were blocked by trastuzumab treatment, suggesting that HER2 is indispensable in MISP phosphorylation (Supplementary Fig. 6e, f). To determine whether MISP phosphorylation is essential for HER2-mediated cell proliferation and trastuzumab sensitivity, we sought to identify potential phosphorylation sites of PLK1 on MISP, using MS analysis. Ten phosphorylation sites of MISP were identified (Supplementary Fig. 5g), and previous studies have shown that three of them were phosphorylated by PLK1 and four by cyclin-dependent kinase 1 (CDK1)[24]. Mutations of three PLK1 phosphorylated residues, S394, S395, and S397, to nonphosphorylatable alanine (MISP-3A) significantly reduced MISP phosphorylation (Fig. 5j). Then, MISP was knocked down by two different shRNAs (Supplementary Fig. 3b) and MISP-WT or MISP-3A were reexpressed into the knocked down cells. Colony formation assay showed that MISP depletion significantly inhibited the cell proliferation and these were reversed by reexpression of MISP-WT instead of MISP-3A (Supplementary Fig. 6h). Also, detection of trastuzumab IC$_{50}$ showed that MISP knockdown significantly sensitized cells to trastuzumab (Fig. 5k). Reexpression of MISP$^{WT}$ into MISP-depleted cells reversed the sensitization effect (IC$_{50}$ = 106.84 μg/mL), while MISP$^{3A}$ reexpression did not (IC$_{50}$ = 40.18 μg/mL, Fig. 5l). Taken together with the above data, we conclude that nuclear SHCBP1 contributes to cell mitosis through binding with PLK1 to promote the phosphorylation of the mitotic interactor MISP, which is the downstream consequence of HER2 activation. HER2 can mediate cell proliferation by activating a Shc1–SHCBP1–PLK1–MISP signaling pathway, which is responsible for trastuzumab sensitivity.

**The mode of SHCBP1–PLK1 interaction.** Having found that HER2 is involved in cell mitotic promotion by activating Shc1–SHCBP1–PLK1–MISP pathway to regulate trastuzumab sensitivity, we posited that pharmacologic inhibitors targeting the SHCBP1–PLK1 may sensitize cells to trastuzumab. To develop a functional inhibitor blocking SHCBP1–PLK1 binding, we sought to identify an SHCBP1 peptide fragment capable of binding to a PLK1 protein. Domain truncation experiments and co-immunoprecipitation revealed that the SHCBP1 355–562 aa residues were required for binding to PLK1 (Fig. 6a). Correspondingly, the domain of PLK1 interacted with SHCBP1 was also determined, and the polo-box domain (PBD), instead of the kinase domain, was found to bind with SHCBP1 (Fig. 6b). This binding was further confirmed by a yeast two-hybrid assay (Supplementary Fig. 7a).

After that, we employed the computational protein–protein docking algorithm (ZDOCK)[25] to predict the bound conformation of the PLK1 PBD domain (PBD ID code 1UMW)[26] and SHCBP1 355–562 aa domain, whose structure was predicted by homologous modeling, using an I-TASSER server[27] (Supplementary Fig. 7b). Following ZDOCK analysis, the top-ranked bound conformation for the SHCBP1–PLK1 complex was selected, and 12 core amino acids on the binding surface of the mode were predicted (Fig. 6c, d and Supplementary Fig. 7c, d). To validate the PLK1–SHCBP1-binding mode, we designed protein deletion mutations and amino acid mutations of PLK1 containing the 12 core amino acids inside the contact surface. A co-immunoprecipitation assay between SHCBP1 and PLK1 mutations showed that mutations containing four core amino acids (K474S, Y485F, H489N, and L490A) significantly blocked the binding of PLK1 and SHCBP1 (Fig. 6e, f). These results confirmed the mode of the SHCBP1–PLK1 complex, in which K474, Y485, H489, and L490 were core amino acids inside the contact surface.

**Natural product TFBG is an inhibitor of SHCBP1–PLK1 interaction.** Given our findings of the SHCBP1–PLK1 interaction mode, we sought to identify small molecules targeting the PLK1–SHCBP1 interaction. By analyzing the SHCBP1–PLK1-binding mode with MOE-Site Finder software, an inhibitor binding pocket of PLK1 (pocket 5) that covered the four core amino acids of SHCBP1–PLK1 complex was found on the binding surface of the complex (Supplementary Fig. 7e). Then, using virtual screening (VS) of diverse chemical libraries consisting of 17,676 small molecules, we initially identified 40 compounds that bind to pocket 5 of PLK1 (Supplementary Fig. 7f and 8a), and the compounds were further screened by surface plasmon resonance (SPR) technology (Supplementary Fig. 8b, c). Finally, a natural product TFBG, which selectively and efficiently targeted PLK1, was identified according to its minimum $K_d$ value (Fig. 6g). SPR kinetic analyses showed that the $K_d$ value of TFBG against PLK1 was $4.67 \times 10^{-7}$ M (Fig. 6h), and the binding affinity was confirmed by a microscale thermophoresis (MST) assay (Supplementary Fig. 8d). To detect the inhibitory effect of TFBG on the SHCBP1–PLK1 interaction, co-immunoprecipitation between SHCBP1 and PLK1 following TFBG exposure indicated that the SHCBP1–PLK1 interaction decreased, with increasing concentrations of TFBG (Supplementary Fig. 8e). FRET assays were conducted and showed that TFBG inhibits the SHCBP1–PLK1 interaction with increasing concentrations and exposure time (Fig. 6i, j). The inhibitory effect of TFBG on the SHCBP1–PLK1 interaction was also supported by a GST pull-down assay, demonstrating that TFBG inhibited the binding of SHCBP1 with PLK1 in vitro (Supplementary Fig. 8f). To

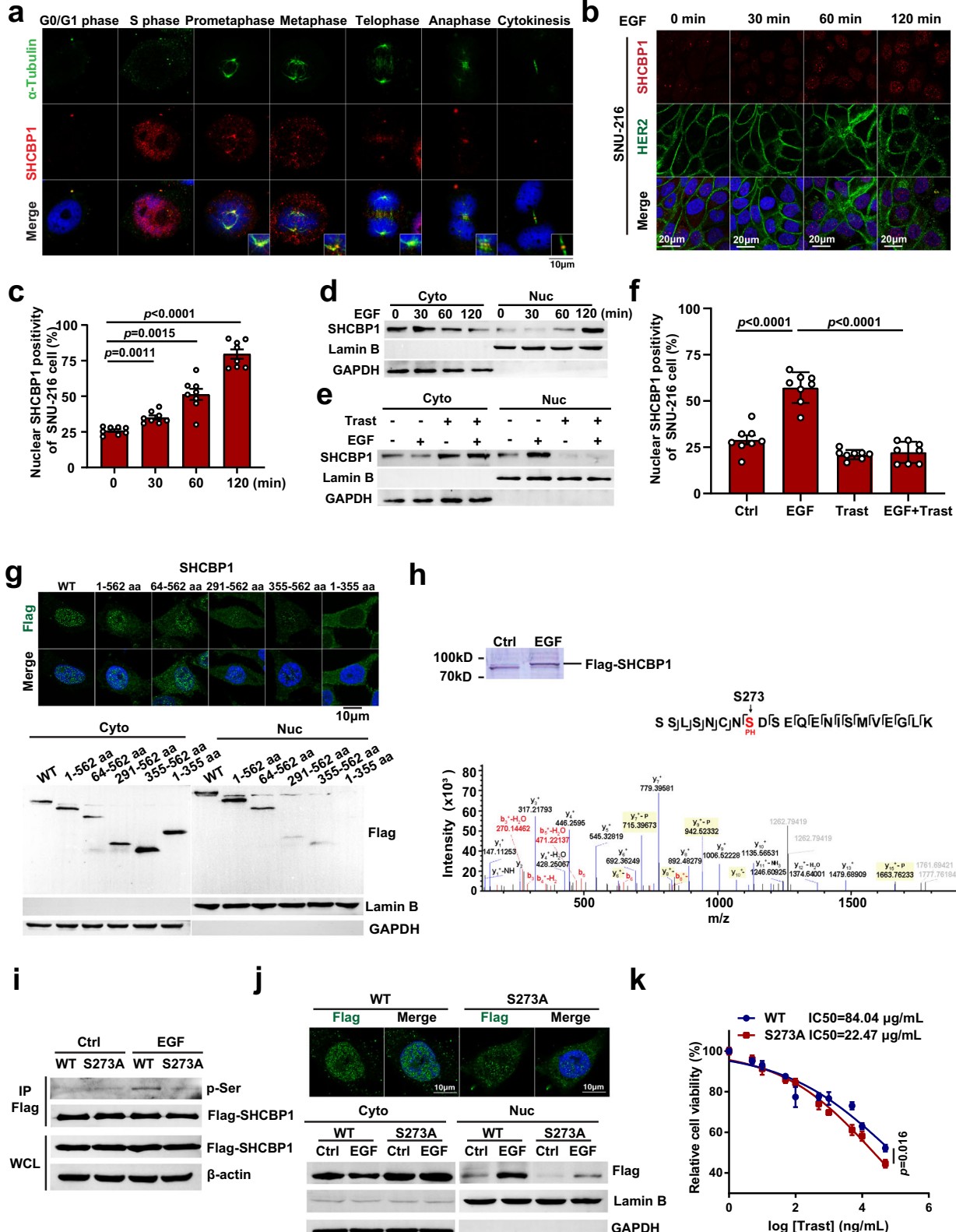

determine the inhibitory effect of TFBG on cell proliferation, we assessed the $IC_{50}$ of the inhibitor, indicating a significant proliferation inhibition both in SNU-216 cells ($IC_{50} = 23.42$ μM, Fig. 6k) and NCI-N87 cells ($IC_{50} = 26.03$ μM, Supplementary Fig. 8g). Taken together, these data demonstrate that TFBG is a selective inhibitor targeting the PLK1–SHCBP1 interaction and displays the potential for use as an anticancer drug.

**TFBG sensitizes gastric cancer to trastuzumab therapy by blocking SHCBP1–PLK1–MISP axis.** To investigate whether combining trastuzumab with inhibitor TFBG might result in additive or synergistic effects on the growth of HER2-positive gastric cancer, we detected the effect of the trastuzumab–TFBG combination on tumor growth both in vitro and in vivo. Cell proliferation assays indicated that the addition of

**Fig. 4 Nuclear localization of SHCBP1 contributes to HER2-mediated cell proliferation. a** Immunofluorescence (IF) staining of SHCBP1 throughout the cell division cycle in synchronized SNU-216 cells. Cells were co-stained with anti-SHCBP1 antibody (red), anti-α-tubulin antibody (green), and DAPI (blue). **b** IF staining of SHCBP1 nuclear localization in SNU-216 cells treated with 100 ng/mL epidermal growth factor (EGF) at the indicated times. Cells were co-stained with anti-SHCBP1 antibody (red), anti-ERBB2 antibody (green), and DAPI (blue). **c** Nuclear SHCBP1 positivity of SNU-216 cells following EGF treatment for different times. Eight fields containing at least 50 nuclei were counted in each treatment group. Data are the mean ± s.e.m. The $p$ values were determined by one-way ANOVA ($n = 8$ independent biological samples). **d** Immunoblotting analysis of SHCBP1 in nuclear (Nuc) and cytosol extracts (Cyto) of EGF-treated SNU-216 cells. **e, f** Immunoblotting analysis (**e**) and IF statistical results (**f**) of nuclear SHCBP1 positivity in SNU-216 cells treated with EGF and/or trastuzumab (Trast). Data are the mean ± s.e.m. The $p$ values were determined by one-way ANOVA ($n = 8$ independent biological samples). **g** Nuclear localization detection of Flag-tagged SHCBP1 deletion mutants stably expressed in SNU-216 cells using IF and immunoblotting. Cells were immunostained with anti-Flag antibody (green) for anti-SHCBP1 deletion mutants and DAPI (blue) for nuclei. WT wild type. **h** Identification of serine residue phosphorylation in SHCBP1. Flag-SHCBP1 was immunopurified and analyzed by SDS–PAGE and LC–MS/MS. **i** Lysates from SNU-216 cells stably expressing Flag-tagged SHCBP1 (WT) or mutant SHCBP1 (S273A) were immunopurified and analyzed by immunoblotting, using an anti-phosphoserine (p-Ser) antibody. IP immunoprecipitation, WCL whole cell lysates. **j** Nuclear localization detection of Flag-tagged SHCBP1 or mutant SHCBP1 (S273A), using IF and immunoblotting. Cells were immunostained with anti-Flag antibody (green) and DAPI (blue). **k** Sensitivity to trastuzumab in SHCBP1 knockdown cells reexpressing SHCBP1 or mutant SHCBP1 (S273A). Cells were treated with trastuzumab at the indicated concentrations for 6 days and subjected to a cell viability assay. Data are the mean ± s.e.m. The $p$ values were determined by two-sided Student's $t$ test ($n = 5$ independent biological samples). Data of **a** and **b** are representative of at least three independent experiments. Data of **d**, **e**, **g**, **i**, and **j** are representative of at least two independent experiments.

TFBG sensitized NCI-N87 and SNU-216 cells to trastuzumab and caused a dose-dependent decrease in cell proliferation (Fig. 7a, b). The addition of 10 μM TFBG resulted in a decrease in trastuzumab $IC_{50}$ from 142.04 to 32.34 μg/mL, and the addition of 20 μM TFBG reduced the $IC_{50}$ to 7.29 μg/mL in SNU-216 cells. Using the Chou and Talalay method, we confirmed that TFBG and trastuzumab in combination exerted a synergistic response in cells with a combination index (CI) < 1 (Supplementary Fig. 9a). A similar combinatorial effect was confirmed by colony formation assays of SUN-216 and NCI-N87 cells (Fig. 7c, d).

For in vivo experiments, we established xenografts of NCI-N87 cells in nude mice and began treatment once tumors reached ~100 mm³ in size with trastuzumab (intraperitoneal, 10 mg/kg, 2×/wk × 3), TFBG (intraperitoneal, 50 mg/kg, 1×/day × 21), or a combination of the agents. We found that both trastuzumab and TFBG monotherapies moderately slowed tumor growth and did not induce regression. However, combination therapy significantly delayed tumor growth more than either single agent and induced dramatic tumor regression (Fig. 7e and Supplementary Fig. 9b–d). Hematoxylin–eosin and immunohistochemical analysis of Ki-67-positive cells in mouse xenograft tumors confirmed that combination therapy reduced the rates of cell proliferation more significantly than either single agent (Fig. 7f). We also treated the NCI-N87 tumors using TFBG subcutaneously at low doses (2.5 mg/kg, 1×/day × 21), confirming that the trastuzumab–TFBG combination suppressed tumor growth efficiently and induced tumor regression dramatically (Supplementary Fig. 9e–g). Finally, we validated whether SHCBP1–PLK1–MISP signaling was blocked in tumors treated by trastuzumab–TFBG combination. SHCBP1 IF staining of tumor tissues demonstrated that TFBG and trastuzumab combination effectively blocked SHCBP1 nuclear translocation (Fig. 7g, h). MISP immunoblotting showed that both trastuzumab and TFBG monotherapies inhibited the MISP phosphorylation and combination therapy significantly enhanced this inhibition (Fig. 7i). To further investigate whether TFBG blocking SHCBP1–PLK1 interaction has feedback effects on the upstream Shc1–SHCBP1 bindings, we detected the Shc1–SHCBP1 interactions after 5 and 10 μM TFBG treatment. We found that TFBG partly suppressed EGF-induced Shc1 and SHCBP1 dissociation, which suggested that the feedback inhibition on Shc1–SHCBP1 of TFBG is one of the reasons why TFBG sensitize gastric cancer to trastuzumab (Supplementary Fig. 9h).

Together, these findings suggested that the combination of TFBG and trastuzumab is efficient in treatment of HER2-positive gastric cancer by blocking HER2–SHCBP1–PLK1–MISP

signaling. TFBG is a promising treatment strategy for sensitizing gastric cancer cells and improving trastuzumab efficacy to battle HER2-positive gastric cancer.

## Discussion

Trastuzumab is one of the approved anti-HER2 therapy in combination with chemotherapy for patients with advanced gastric cancer, but its antitumor efficacies are less favorable due to insufficient cell sensitivity and drug resistance[28,29]. The mechanisms underlying drug resistance are usually proposed to be amplification, upregulation, or mutation of HER2 and downstream MAPK and PI3K/AKT pathways[7,8]. Here, we elaborated a new mechanism that HER2 promotes tumorigenesis by direct regulation of cell mitotic progression through hyper-activating a Shc1–SHCBP1–PLK1–MISP axis, which drives the sensitivity of HER2-positive cells to trastuzumab. The Inhibitor TFBG can selectively block SHCBP1–PLK1 complex and render gastric cancer sensitive to trastuzumab (Fig. 8). Our findings facilitate a deeper understanding of HER2-evoked intracellular signaling networks and offer a promising therapeutic strategy for HER2-positive gastric cancer.

HER2 employs a famous scaffold protein, Shc1, to amplify downstream signaling through multiple waves of protein interactions[21,30]. We screened out a Shc1-binding protein, SHCBP1, which was upregulated and associated with worse OS and clinicopathological characteristics in HER2-positive patients. Previous studies also demonstrated the oncogenic role of SHCBP1 in non-small cell lung carcinoma (NSCLC) and breast cancer. It was found that SHCBP1 was upregulated and correlated with poorer survival of patients with NSCLC and breast cancer[31,32]. Moreover, SHCBP1 was found strongly associated with the HER2 expression in breast cancer[32]. In here, we demonstrated a positive correlation between HER2 and SHCBP1 expression in gastric cancer, which suggested that the SHCBP1 may be involved in HER2-promoted tumorigenesis. We knocked down the SHCBP1 expression and found that SHCBP1 depletion inhibited the cell proliferation, and sensitized the cell to trastuzumab in gastric cancer cell SNU-216 and NCI-N87. Consistently, others also reported that SHCBP1 knockdown inhibited the proliferation and metastasis of gastric cancer cell MGC-803 and SGC-7901, and suppressed the proliferation and motility of esophageal squamous cell carcinoma ESCC cells[33,34]. These findings demonstrated the oncogenic role of SHCBP1 in a variety of human cancers. Previously, SHCBP1 was identified to play a role in cisplatin-induced apoptosis resistance in NSCLC, indicating the regulatory function of SHCBP1 in drug resistance to

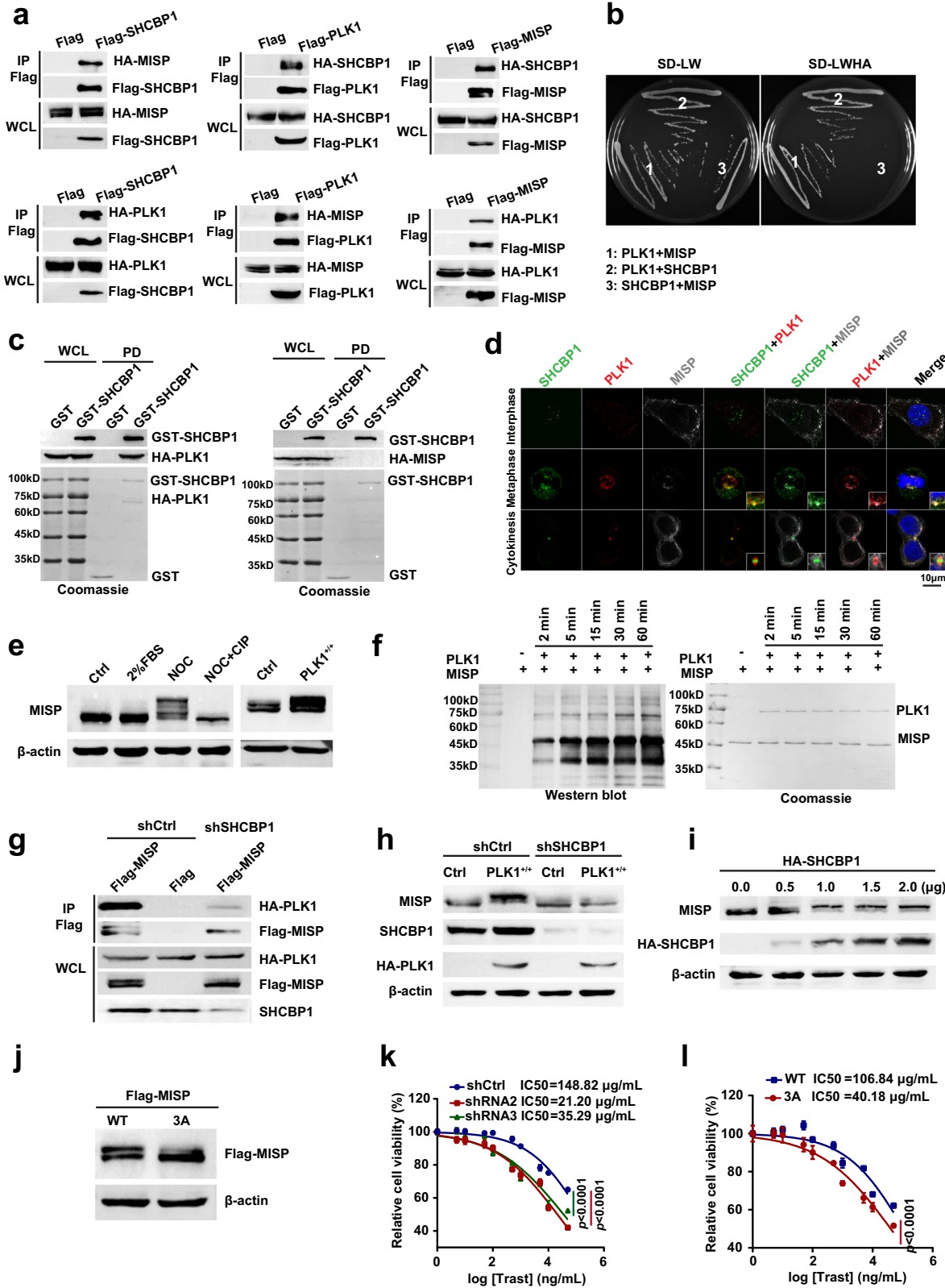

cancers[35]. Notably, we showed that SHCBP1 functions as a regulator of HER2-directed gastric cancer therapy. SHCBP1 was found correlating with trastuzumab sensitivity in patients who received trastuzumab-based therapy, and SHCBP1 deletion rendered gastric cancer sensitive to trastuzumab, implying that SHCBP1 contributed to trastuzumab sensitivity and was targeted therapeutically. However, more clinical analysis of correlation

between SHCBP1 and HER2-positive patients who received trastuzumab therapy have not been conducted in here due to the shortage of the patients. A large-scale clinical study is required in the future to validate the importance of SHCBP1 in trastuzumab-based therapy.

Previously, SHCBP1 was reported as an Shc1-binding protein[21,36]. We showed that SHCBP1 was initially associated

**Fig. 5 SHCBP1 binds with PLK1 to promote MISP phosphorylation for mitotic progression. a** Co-immunoprecipitation assays of Flag/HA-tagged SHCBP1, PLK1, or MISP co-expressed in HEK293T cells, which were synchronized to the mitotic phase using 40 ng/mL nocodazole (NOC). IP immunoprecipitation, WCL whole cell lysates. **b** Yeast two-hybrid interaction assays of the SHCBP1, PLK1, and MISP complex. SD-LW synthetic dropout Leu and Trp, SD-LWHA synthetic dropout Leu, Trp, His, and adenine. **c** Pulldown assays of the SHCBP1, PLK1, and MISP complex. 6His-GST-tagged SHCBP1 and HA-tagged PLK1 or MISP expressed in *E. coli* were pulled down and analyzed by Coomassie blue staining and immunoblotting. WCL whole cell lysates, PD pulldown. **d** Colocalization of SHCBP1, PLK1, and MISP in SNU-216 cells. Cells stably expressing GFP-tagged SHCBP1 (green) were immunostained with anti-PLK1 antibody (red), anti-MISP antibody (gray), and DAPI (blue). **e** Immunoblotting assays of MISP mobility shift in nocodazole (NOC) blocked SNU-216 cells with or without calf intestinal phosphatase (CIP) treatment and PLK1-overexpressed SNU-216 cells. **f** In vitro kinase assays of PLK1 on MISP detected by immunoblotting (left) and Coomassie blue staining (right). **g** Co-immunoprecipitation assays of Flag-MISP and HA-PLK1 in SNU-216 shCtrl or SHCBP1 knockdown cells. **h** Detection of overexpressed PLK1-induced MISP mobility shift in SNU-216 shCtrl or SHCBP1 knockdown cells, using immunoblotting. **i** Detection of MISP mobility shift in SHCBP1 knockdown cells reexpressing SHCBP1 at the indicated concentrations using immunoblotting. **j** Detection of MISP mobility shift in cells transfected with Flag-tagged MISP (WT) or mutant MISP (3A: S394A, S395A, and S397A) by immunoblotting using anti-Flag antibody. **k** Sensitivity to trastuzumab in SNU-216 shCtrl cells or MISP knockdown cells. Each group was treated with trastuzumab for 6 days and subjected to a cell viability assay. Data are the mean ± s.e.m. The *p* values were determined by one-way ANOVA (*n* = 6 independent biological samples). **l** Sensitivity to trastuzumab in MISP knockdown SNU-216 cells reexpressing MISP (WT) or mutant MISP (3A: S394A, S395A, and S397A). Cell was treated with trastuzumab for 6 days and subjected to a cell viability assay. WT wild type. Data are the mean ± s.e.m. The *p* values were determined by two-sided Student's *t* test (*n* = 6 independent biological samples). Data of **a**, **c**, **e**–**j** are representative of two independent experiments. Data of **b** and **d** are representative of three independent experiments.

with Shc1, but displaced following EGF stimulation and that this was a downstream consequence of HER2 activation. Despite the detailed mechanism of how HER2 activation disrupts SHCBP1–Shc1 binding has not been uncovered in here, it has been reported that SHCBP1 interacted with the SH2 domain of Shc1, which is also recruited to phosphotyrosine motifs on ERBBs[36,37]. SHCBP1 and ERBBs bind competitively to the Shc1 SH2 domain, which is a reasonable explanation for HER2-induced SHCBP1–Shc1 dissociation, but a systematic study is required in the future. It was reported that the silencing of SHCBP1 led to an increase of PTEN in lung cancer[38]. It is possible that SHCBP1 synergistically activates the PI3K/AKT signaling by inhibiting PTEN. Revealing the mechanism of the crosstalk between HER2/SHCBP1/PLK1 and PI3K/AKT or MAPK signaling in the future will be valuable for the understanding of SHCBP1 as HER2 downstream effector.

Following the disconnection of SHCBP1 and Shc1, we demonstrated that SHCBP1 translocated into the nucleus to promote cell mitotic progression. Dramatic nuclear localization of SHCBP1 after EGF exposure was observed and abrogated by trastuzumab treatment. We also revealed that blocking SHCBP1 nuclear localization rendered gastric cancer cells sensitive to trastuzumab, demonstrating that SHCBP1 nuclear localization is a downstream consequence of HER2 activation. However, it is unknown how SHCBP1 translocates into the nucleus following dissociation from Shc1. We characterized that EGF-induced S273 phosphorylation of SHCBP1 was responsible for these effects. Interestingly, previous observations also indicated that EGF stimulation can cause SHCBP1 redistribution to the nucleus. SHCBP1 interacts with β-catenin for nuclear translocation to activate β-catenin signaling[31]. Consistent with this possibility, we speculate that phosphorylation at S273 causes the release of SHCBP1 from Shc1 and a delayed interaction with β-catenin, resulting in the nuclear translocation of SHCBP1. However, a detailed interrogation is needed for the explanation.

Mitotic regulators always translocate into the nucleus for cell division before the nuclear membrane cracking[39], and we found that SHCBP1 is an essential mitotic regulator involved in mitotic progression following its nuclear localization. SHCBP1 binds with PLK1 and forms a complex with MISP to regulate cell mitosis through colocalization at the spindle poles. PLK1 is an essential polo-like kinase and performs pleiotropic functions to control mitotic entry, centrosome separation, and cytokinesis[40,41]. MISP is a substrate of PLK1 and is phosphorylated by PLK1 to stabilize cortical and astral microtubule attachments required for proper

mitotic spindle positioning[24,42]. We showed that nuclear SHCBP1 contributed to cell mitosis through binding with PLK1 to promote the phosphorylation of MISP, which was a downstream consequence of HER2 activation and involved in HER2-mediated cell proliferation and trastuzumab sensitivity. We found that SHCBP1 localized to the spindle poles in metaphase and to the midbody during cytokinesis, suggesting that SHCBP1 plays pleiotropic roles in both metaphase and cytokinesis. Indeed, SHCBP1 was reported to be responsible for cell mitosis during cytokinesis, which was colocalized at the midbody with MgcRacGAP and MKLP1, where it forms centralspindlin and promotes the ingression of the cytokinetic furrow[22,23]. Our findings provide a deeper understanding of SHCBP1 as a mitotic protein employed in a plethora of mitotic processes.

From a clinical perspective, our findings support that SHCBP1–PLK1–MISP signaling serves as a HER2 downstream signaling pathway to render gastric cancer sensitive to trastuzumab. To this end, we identified a natural product TFBG, as an inhibitor to block the SHCBP1–PLK1 interaction and sensitize cells to trastuzumab. TFBG is a polyphenolic compound extracted from black tea and has exhibited potent anticancer properties[43]. We showed that trastuzumab–TFBG combination therapy significantly delayed tumor growth and induced dramatic tumor regression by blocking activation of SHCBP1–PLK1–MISP signaling, demonstrating that TFBG is a promising sensitizing agent for combination with trastuzumab to battle HER2-positive gastric cancer. However, validation of the tumor inhibition using patient-derived xenografts mouse, and systematic toxicological testing of trastuzumab and TFBG combination is necessary in the future for the developing TFBG as drugs. It should be noted that TFBG is screened out based on the pocket of PLK1, which interacts with SHCBP1 to block SHCBP1–PLK1 interaction. It is possible that TFBG may inhibit the binding of PLK1 with other proteins. Previous studies revealed that TFBG also target to other signal pathways including EGFR[44], which may partly be responsible for the sensitization effects of TFBG to trastuzumab in HER2-positive gastric cancer. In addition, some studies underlined the relatively low bioavailability and metabolic stability of TFBG[45]. TFBG is extensively metabolized to metabolites, including theaflavin, theaflavin-3-gallate, and gallic acid[46]. Thus, medicinal chemistry efforts aimed at improving the compound's pharmacological properties to the level required for drugs will be important.

In conclusion, we propose that HER2 is involved in cell mitotic regulation by hyperactivating an intracellular SHCBP1–PLK1–MISP

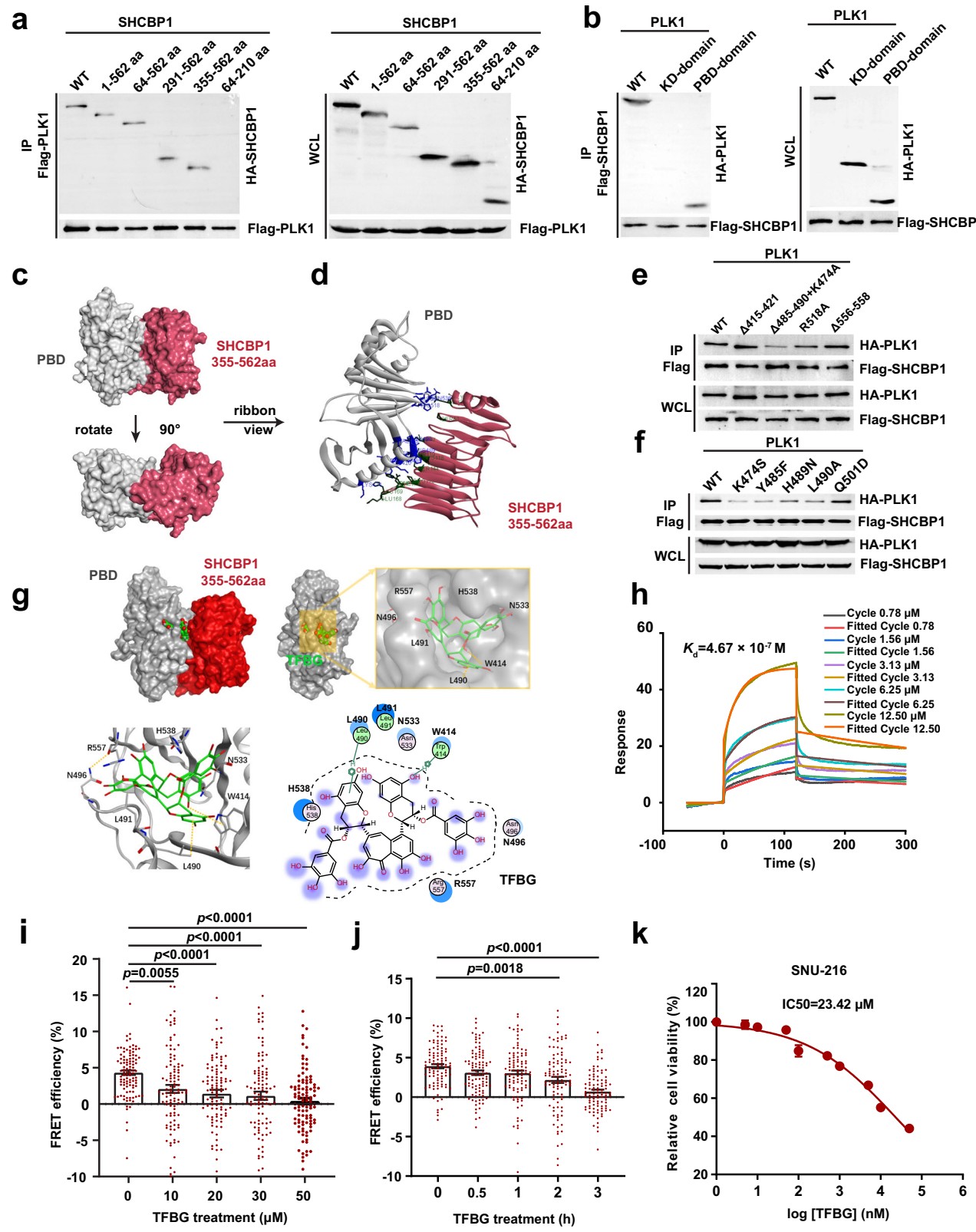

signaling to mediate cell proliferation and drug sensitivity. HER2 activation leads to the recruitment of Shc1 to activate the MAPK and PI3K/AKT pathways and in turn releases SHCBP1, which translocates to the nucleus for mitotic progression. The SHCBP1–PLK1–MISP axis is an essential intracellular cascade of HER2 activation in addition to the canonical downstream signaling pathways. Blocking SHCBP1–PLK1 complex using targeted TFBG

agent renders gastric cancer sensitive to HER2-directed therapy, providing additional combination therapy in HER2-positive patients with advanced gastric cancer.

## Methods

**Human gastric cancer clinical specimens.** All specimens were acquired from patients under the auspices of clinical protocols approved by the Medical Ethics

**Fig. 6 Natural product theaflavine-3, 3′-digallate (TFBG) blocks the SHCBP1–PLK1 interaction. a** Interactions between PLK1 and different SHCBP1 deletion mutants analyzed by co-immunoprecipitation assays. WT wild type, IP immunoprecipitation, WCL whole cell lysates. **b** Interactions between SHCBP1 and the kinase domain (KD) or polo-box domain (PBD) of PLK1 analyzed by co-immunoprecipitation assays. **c** The bound conformation of the SHCBP1 355–562 aa and PLK1 PBD as predicted by the ZDOCK algorithm. The PBD is displayed in gray, and the SHCBP1 355–562 aa domain is displayed in red. **d** The protein complex of PBD and SHCBP1 355–562 aa displayed in ribbon mode from the same angle view as in **c**. The PBD residue bonds on the binding surface are marked in blue. **e** Interactions between SHCBP1 and different PLK1 deletion mutants containing residues on the binding surface of the mode analyzed by co-immunoprecipitation assays. **f** Interactions between SHCBP1 and different PLK1 mutants of four core amino acids on the binding surface of the mode analyzed by co-immunoprecipitation assays. **g** Computational model and interactions of TFBG and the SHCBP1–PLK1 complex (top left and right). Overall structure of the complex showing the binding of TFBG to PLK1 and SHCBP1, which are in green, gray, and red (bottom left and right). Architecture of PLK1 and TFBG showing interacting amino acids of PLK1 (gray) and TFBG (green). **h** Kinetic constant ($K_d$) analyses of TFBG interacting with PLK1 using surface plasmon resonance (SPR) assays. **i, j** Fluorescence resonance energy transfer (FRET) analysis of eCFP-PLK1 and eYFP-SHCBP1 in cells treated with TFBG at the indicated concentrations (**i**, $n = 104$ independent cells per group) and time (**j**, $n = 100$ independent cells per group). Data are the mean ± s.e.m. The p values were determined by one-way ANOVA and repeated measures one-way ANOVA. **k** TFBG dose-dependent response in SNU-216 cells for 4 days. Data are the mean ± s.e.m ($n = 6$ independent biological samples). Data of **a**, **b**, **e**, and **f** are representative of two independent experiments.

---

Review Board at the Lanzhou University Second Hospital. Informed consent was obtained from all participants. A total of 223 paired gastric cancer and adjacent normal tissues were constructed to TMAs after formalin-fixed and paraffin-embedded blocks, and prepared for immunohistochemical staining. A total of 16 paired gastric cancer and adjacent normal tissues were used for microarray assay and 8 paired tissues were used for immunoblotting. A total of 22 specimens of HER2-positive gastric cancer patients who received trastuzumab-based therapy were used for immunohistochemical staining to demonstrate the correlation of SHCBP1 with trastuzumab sensitivity (Supplementary Table 2). A consent to publish clinical information reported in Supplementary Fig. 2 and potentially identifying individuals was obtained.

**Cell lines, cell culture, and lentiviral production**. Human gastric cancer cells NCI-N87, HGC-27, MKN-45, and AGS were obtained from the institute of Basic Medical Sciences, Chinese Academy of Medical Sciences (Beijing, China). Human gastric cancer cells KATO-II, and HS-746T were obtained from the Kunming institute of zoology, Chinese Academy of Sciences (Kunming, China). HEK293T cells were obtained from the ATCC, and human gastric cancer cells SNU-216 were obtained from the Korean Cell Line Bank. All cell lines have been tested for mycoplasma contamination and were validated by short tandem repeat DNA fingerprinting, using the commercially available EX20 Kit from AGCU. NCI-N87 and SNU-216 cells were grown in RPMI medium supplemented with 10% fetal bovine serum (FBS) and other cell lines were cultured in DMEM/F12 medium supplemented with 10% FBS. All cell lines were incubated at 37 °C with 5% $CO_2$ under humidifying conditions.

To generate SNU-216 stable cell lines expressing Flag-tagged Shc1, SHCBP1, or MISP, HEK293T cells were used to package virus. Pseudotyped virus was produced by co-transfecting 1 µg LentiCMV-FLAG-Shc1, SHCBP1, or MISP, 1 µg pDD and 0.5 µg pVSVg in a 3.5 cm dish of 293T cells. Cell culture supernatants containing recombinant lentiviruses were collected 48 h after transfection and filtered through non-pyrogenic filters with a pore size of 0.45 mm. Supernatants were used at a dilution of 1:2 immediately to transduce SNU-216 cells in the presence of polybrene (10 mg/mL). Transduced cells were selected with 2 µg/mL puromycin for 5–7 days to generate SNU-216 stable lines expressing Flag-tagged Shc1, SHCBP1, or MISP.

**Plasmids and RNA interference**. All the plasmids used in this paper were generated using the Gibson assembly cloning method. The cDNA sequences of *Shc1*, *SHCBP1*, *PLK1*, and *MISP* gene (Genechem, China) were delivered into pLentiCMV-1 × Flag-puro vector (gift from Hui Sun Lab) for conditional expression, into pRK5-vector (gift from Hui Sun Lab) for temporary expression. The mutant constructs of SHCBP1 (S273A and S273D), MISP (S394A, S395A and S397A), and PLK1 (K474S, Y485F, H489N, L490A, and Q501D) were generated by site-directed mutagenesis and cloned into pLentiCMV-1 × Flag-puro or pRK5-vector. The *Shc1* and *PLK1* sequences were cloned into pECFP-N1 and *SHCBP1* to pEYFP-N1 vector (Addgene) for FRET. The *SHCBP1*, *MISP*, and *PLK1* were cloned into pGBKT7 and pACT2 vector (Addgene) for yeast two-hybrid assay, and cloned into pETDuetM-6HIS-GST-pp (gift from Sanduo Zheng Lab) for GST pulldown assay. All constructs were verified by full-length sequencing. The lentiviral vectors encoding human *SHCBP1*, *MISP*, and scrambled control shRNA were obtained from Genechem company. The lentiviral vector containing shRNA inserts was packaged by co-transfection into HEK293T with pSPAX2 and pMD2G. SNU-216 and NCI-N87 cells were infected with lentivirus and selected with puromycin. Knockdown of SHCBP1 or MISP expression were validated by quantitative RT-PCR and immunoblotting. The information of shRNA sequences was shown in Supplementary Table 3.

**Immunohistochemistry**. TMAs of human gastric cancer specimens were deparaffinized and rehydrated, followed by antigen retrieval. After anti-SHCBP1 (Sigma, diluted at 1:200) or anti-ERBB2 (Abcam, diluted at 1:200) antibody incubation, the slides were dehydrated and stabilized with mounting medium and the images were acquired by a KF-PRO-120 scanner (Konfoong, China). Staining intensity (0, 1, 2, and 3) and percentage of positive cells among cancer duct (0–100%) were evaluated by a pathologist at the Lanzhou University Second Hospital. The final histoscore (H-score) were calculated by multiplying the staining intensity and the percentage of positive cells. The H-score < 70 is divided as SHCBP1 low expression group and H-score ≥ 70 as SHCBP1 high expression group, according to the median value of SHCBP1 H-score.

**Confocal microscopy analysis**. TMAs or cells cultured on 35 mm glass-bottomed Microwell Dishes (MatTek Corporation) were fixed with 4% paraformaldehyde for 10 min and subjected to permeabilization with 0.1% TritonX-100. After that, cells were incubated with 3% bovine serum for 1 h at room temperature and then with primary antibodies overnight at 4 °C, washed thrice in PBS and further incubated with the appropriate fluorescent-labeled secondary antibodies. Nuclei were counterstained with 4, 6-diamidino-2-phenylindole (DAPI) before mounting. Confocal fluorescence images were captured using a Zeiss LSM 880 laser microscope (×63 oil objective, Plan-Apochrom 1.4). The positivity of SHCBP1 in the nucleus was detected by SHCBP1 IF and was quantified using Image J V1.53c software. The mean fluorescence intensity (MFI) of SHCBP1 in the nucleus was quantified and the MFI > 10 was identified as the positive cells.

**Time-lapse and flow cytometry analysis**. For the time-lapse analysis, the shCtrl and SHCBP1 knockdown cells were cultured on glass-bottom dishes. Starting 24 h later, the cells were monitored using a time-lapse microscope system (Operetta CLS, PerkinElmer) for 24 h and the images were captured every 20 min. For cell cycle detection, the shCtrl and SHCBP1 knockdown cells were collected and fixed with precooled 75% ethanol overnight at 4 °C. The cells were submersed in 37 °C water for 30 min, followed by washing with PBS buffer twice and blocked in PI/RNase staining buffer (BD) for 15 min at room temperature before analysis. The DNA content of the cell cycle was measured by flow cytometer (CytoFLEX, Beckman Coulter).

**Immunoblotting**. Cells were lysed with RIPA buffer and centrifuged at $13,800 \times g$ for 10 min. Protein lysates were separated by SDS/PAGE, blotted onto a PVDF membrane, and probed overnight at 4 °C with antibodies against rabbit anti-SHCBP1 (1:1000; Sigma), mouse anti-HER2 (1:1000, Abcam), mouse anti-Flag (1:1000, Sigma), mouse anti-HA (1:1000, Invitrogen), rabbit anti-MISP (1:1000, Abcam), rabbit anti-PLK1 (1:1000, Proteintech), rabbit anti-Shc (1:1000, Abcam), mouse anti-β-actin (1:1000, Proteintech), mouse anti-GAPDH (1:1000, Proteintech) or rabbit anti-LaminB (1:1000, Proteintech), rabbit anti-thiophosphate ester antibody (1:5000, Abcam), and rabbit anti-phosphoserine (1:500, Abcam). Stripped membranes were probed with a secondary antibody of goat anti-mouse or anti-rabbit IgG (1:10,000, Bioworld) and then visualized with enhanced chemiluminescence.

**Co-immunoprecipitation assay**. Cells seeded in 10 cm dishes were lysed with lysis buffer [50 mM Tris-HCl (pH = 7.4), 150 mM NaCl, 1 mM EDTA, and 0.2% TritonX-100] containing protease inhibitor. After 10 min on ice, lysates were centrifuged at $13,800 \times g$ for 10 min at 4 °C. The supernatants were incubated with 20 µL prewashed anti-Flag M2 affinity gel agarose (50% slurry, Sigma) for 2 h on a rotary shaker at 4 °C. Immunoprecipitates were collected by centrifugation at $1500 \times g$ for 2 min at 4 °C, washed thrice with 1 mL of cold lysis buffer, and eluted

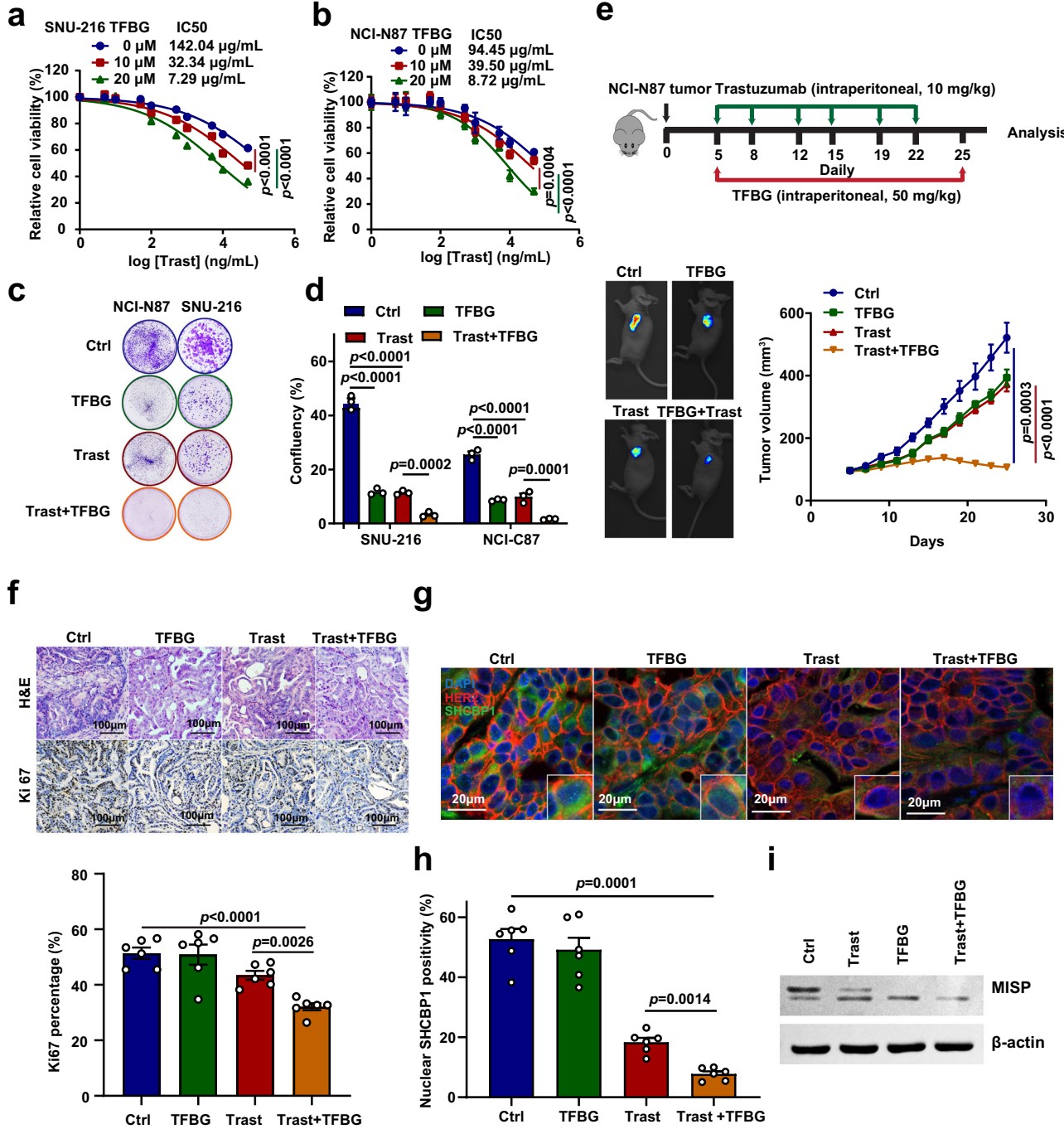

**Fig. 7 Trastuzumab–TFBG combination is efficient in treatment of HER2-positive gastric cancer. a, b** Sensitivity to trastuzumab (Trast) in SNU-216 (**a**) and NCI-N87 (**b**) cells treated with theaflavine-3, 3'-digallate (TFBG) at the indicated concentrations. Cell proliferation was analyzed after 6 days of treatment. Data are the mean ± s.e.m. The *p* values were determined by one-way ANOVA (*n* = 6 independent biological samples). **c, d** Colony formation (**c**) and the statistical results (**d**) of NCI-N87 and SNU-216 cells in the continuous presence of trastuzumab (10 μg/mL) and/or TFBG (10 μM). Data are the mean ± s.e.m. The *p* values were determined by one-way ANOVA (*n* = 3 independent biological samples). **e** Representative image and tumor growth curves of xenograft mice carrying NCI-N87 cell tumors treated with trastuzumab (intraperitoneal, 10 mg/kg, 2×/wk × 3), TFBG (intraperitoneal, 50 mg/kg, 1×/day × 21), or a combination of the agents. The detailed strategy of drugs treatment is displayed (**e**, top). Data are the mean ± s.e.m. The *p* values were determined by one-way ANOVA (*n* = 8 independent mice per group). **f** Representative images of hematoxylin–eosin (H&E) and Ki-67 immunohistochemical analysis (top) and quantification of Ki-67-positive cells (bottom) in mice xenograft tumors treated with trastuzumab and/or TFBG. Data are the mean ± s.e.m. The *p* values were determined by one-way ANOVA (*n* = 6 independent biological samples). **g, h** Immunofluorescence staining and the statistical results of SHCBP1 nuclear localization in mice xenograft tumors. Tissues were co-stained with anti-SHCBP1 antibody (green), anti-ERBB2 antibody (red), and DAPI (blue). The positivity of SHCBP1 in the nucleus was quantified using Image J V1.53c software. Data are the mean ± s.e.m. The *p* values were determined by one-way ANOVA (*n* = 6 independent biological samples). **i** Detection of MISP phosphorylation in mice xenograft tumors by immunoblotting. Data of **i** are representative of two independent experiments.

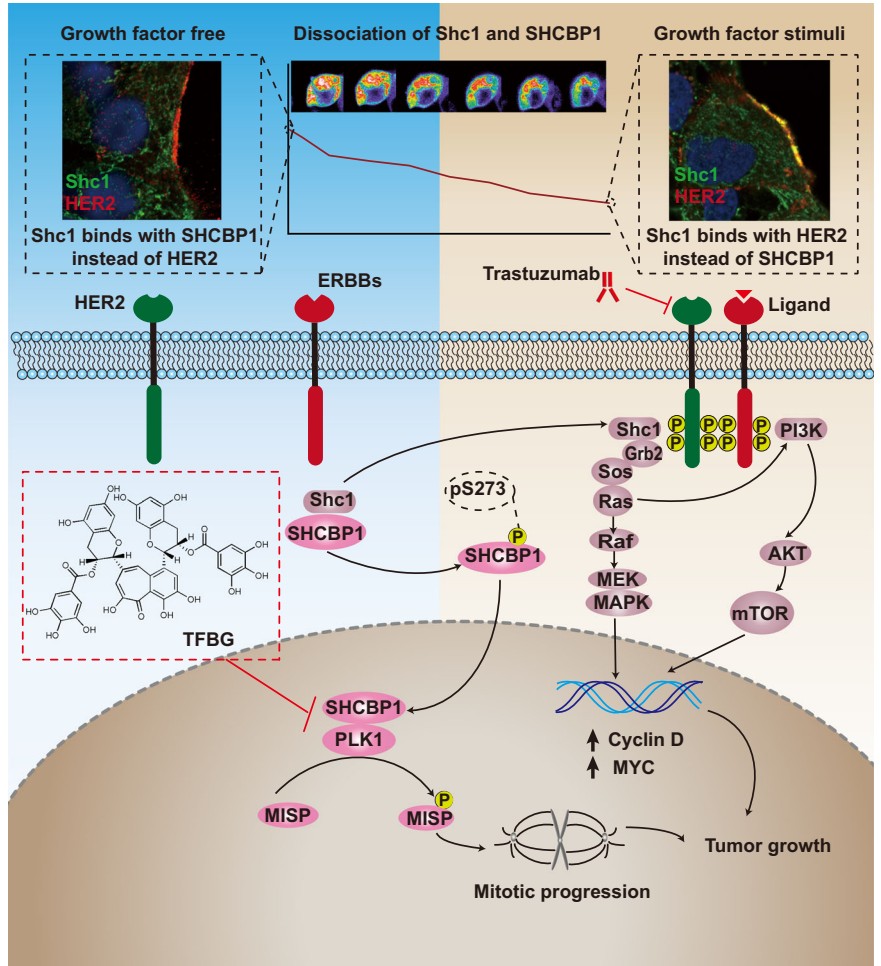

**Fig. 8 HER2 promotes tumorigenesis by regulating cell mitotic progression through Shc1–SHCBP1–PLK1–MISP signaling.** In growth factor free condition, HER2 is inactivated and SHCBP1 acts as a Shc1-binding protein interacting with Shc1. Following growth factor stimuli, HER2 is activated through dimerization with other HER members and recruits Shc1 to evoke MAPK or PI3K pathways. Released SHCBP1 responds to HER2 cascade by translocating into the nucleus following Ser273 phosphorylation (pS273), and then contributing to cell mitosis regulation through binding with PLK1 to promote the phosphorylation of the mitotic interactor MISP. Hyperactivation of HER2–SHCBP1–PLK1 axis impairs the sensitivity of HER2-targeted therapy trastuzumab. The Inhibitor theaflavine-3, 3'-digallate (TFBG) blocks SHCBP1–PLK1 complex and renders gastric cancer sensitive to trastuzumab. TFBG–trastuzumab combination is highly efficacious to suppress gastric cancer growth.

by adding 0.1 M glycine HCl (pH 3.5). Finally, the protein samples were analyzed by SDS–PAGE and immunoblotted with antibodies against HA and Flag.

**Affinity purification and mass spectrometry analysis**. SNU-216 stable lines expressing Flag-epitope-tagged Shc1, SHCBP1, or MISP were seeded in 10 cm dishes and treated as indicated in the figure legends. Cells were lysed with lysis buffer [50 mM Tris-HCl (pH = 7.4), 150 mM NaCl, 1 mM EDTA, and 0.2% Tri-tonX-100] containing protease inhibitor. Lysates were centrifuged at $13,800 \times g$ for 10 min at 4 °C. The supernatants were incubated with 50 μL prewashed anti-Flag M2 affinity gel agarose (50% slurry, Sigma) for 2 h on a rotary shaker at 4 °C. Immunoprecipitates were collected by centrifugation at $1500 \times g$ for 2 min at 4 °C, washed thrice with 1 mL of cold lysis buffer, and eluted by adding 0.1 M glycine HCl (pH = 3.5). To identify interacting proteins of Shc1 or SHCBP1, the immune-complex samples were subjected to in-gel trypsin digestion overnight, and the extracted peptides subjected to LC–MS/MS analysis. To identify the phosphor-ylation sites of SHCBP1 and MISP, samples were run on SDS–PAGE, and the position of the differentially phosphorylated bands was determined by conducting western blot analysis on an aliquot of the same samples. Bands were then excised and subjected to LC–MS/MS analysis.

**Yeast two-hybrid assay**. The yeast two-hybrid assay was conducted by the Matchmaker Yeast Two-Hybrid System (Clontech), according to the manu-facturer's instruction. In brief, *SHCBP1* or *MISP* were cloned into the pGBKT7 vector downstream of a Gal4 DNA-binding domain to construct a bait plasmid, and *SHCBP1* or *PLK1* were cloned into the pACT2 vector downstream of a Gal4 DNA-activation domain to construct a prey plasmid. The bait and prey plasmids

were transformed into Y187 yeast and the resulting diploids were grown on -Leu/-Trp, and subsequently assayed for the activation of reporter genes [growth on quadruple dropout medium (-Ade/-His/-Leu/-Trp)].

**GST pulldown assay**. *SHCBP1*, *PLK1*, and *MISP* gene were cloned into the pETDuetM-6HIS-GST-pp vector. GST-tagged SHCBP1, His-tagged PLK1, and His-tagged MISP proteins were expressed in the *Escherichia coli* strain BL21 (DE3) strain and GST pulldown assay was conducted, using a pierce GST protein interaction pulldown kit (Thermo Scientific). Input and eluate samples were resolved using SDS–PAGE and analyzed by Coomassie blue staining or immunoblotting.

**Cell viability assay and colony formation assay**. For cell viability studies, $5 \times 10^3$ cells were plated in 96-well plates and incubated overnight for cell attachment. Cells were treated for 6 days with indicated concentrations of trastuzumab, TFBG, or in combination. Cell viability was measured using the CellTiter 96A Queous One Solution Cell Proliferation Assay (Promega), according to manufacturer's instructions. Plates were read at 490 nm absorbance. The IC$_{50}$ was analyzed by the GraphPad Prism 8.3 software. The combined drug effects of trastuzumab and TFBG were analyzed using CalcuSyn V2.0 software developed by Chou and Talalay[47]. For colony forming assay, 1000 cells were seeded in 35 mm dishes and treated continuously with trastuzumab, TFBG, or in combination. The colonies were fixed in 4% formaldehyde, stained with 0.05% (w/v) crystal violet, and counted by Image J V1.53c software after 10 days.

**RT-PCR**. Total RNA was extracted from harvested cells with Trizol reagent, and reversely transcribed into cDNA with the reverse transcription kit, according to manufacturer's protocols. Quantitative RT-PCR amplification and product detection were conducted on a LightCycler instrument (Roche) by using SYBR Green dye (Takara) and 10 μM forward and reverse primers. The information of primer sequences used for qRT-PCR were revealed in Supplementary Table 4.

**In vitro phosphorylation assay**. A total of 100 ng PLK1 and 500 ng MISP proteins (330–679 aa) were incubated with 1 mM γ-[(6-azidohexyl) imido] ATP (Abcam) in 30 μL kinase buffer (25 mM Tris-HCl, pH 7.5, 5 mM β-glycerophosphate, 2 mM DTT, 0.1 mM Na$_3$VO$_4$, and 10 mM MgCl$_2$) for the indicated times at 37 °C, and then alkylated with 2.5 mM p-nitrobenzyl mesylate (Abcam) for 2 h at 37 °C. The samples were resolved by 12% SDS–PAGE, and subjected to coomassie blue staining and immunoblotting with thiophosphate ester antibody.

**Fluorescence resonance energy transfer assay**. FRET measurement was performed using the acceptor-photobleaching method. For PLK1–SHCBP1 interaction analysis, cells were transfected with eYFP-SHCBP1 and eCFP-PLK1. For Shc1–SHCBP1-binding detection, cells were transfected with eYFP-SHCBP1 and eCFP-Shc1. Cells were seeded on 35 mm glass-bottomed microwell dishes (MatTek Corporation) and imaging analyses were carried out using a Zeiss LSM 880 laser microscope, a ×63 oil objective (Plan-Apochrom 1.4). Intensity of eYFP and eCFP was collected using appropriate bandpass filter (eCFP: Ex 440 nm, Em 563–520 nm; eYFP: Ex 514 nm, Em 520–620 nm). eYFP was bleached with intense 514 nm laser and average intensity of eCFP in a region of interest spanning the bleached cell was determined in the images before acceptor bleach (I1) and after acceptor bleach (I2). FRET efficiency is reported as [1 − (I1/I2)]%. Calculations were based on more than 100 cell images.

**Xenograft studies**. Animals were manipulated and housed according to protocols approved by the Animal Ethics Committee of Lanzhou University Second Hospital. All animals received humane care according to the criteria outlined in the "Guide for the Care and Use of Laboratory Animals" prepared by the National Academy of Sciences and published by the National Institutes of Health. NCI-N87/luc tumor xenograft models were established in athymic nude mice. Tumors were imaged by an in vivo Imaging System (Vieworks, Smart-LF, Korean) on the days shown. To evaluate SHCBP1 knockdown involving in the gastric cancer sensitivity to trastuzumab, female BALB/c nude mice (6–8 weeks of age) received single subcutaneous flank injection of 1 × 10$^6$ Lenti-vector or Lenti-shRNA targeting NCI-N87/luc cells in a 1:1 PBS:Matrigel suspension (BD Biosystems, San Jose, CA). When tumor volumes reached 100 mm$^3$, trastuzumab was given at 10 mg/kg intraperitoneally twice weekly. Tumors were measured by caliper three times per week. Animals were sacrificed and immunohistochemical staining was performed in the tumors after drug treatment for 21 days. To determine the synergistic effects of combining trastuzumab with TFBG, NCI-N87/luc cells (1 × 10$^6$ cells per site) in suspension were mixed with equal volumes of Matrigel and injected subcutaneously into 6–8-week old female BALB/c nude mice. When tumor volumes reached 100 mm$^3$, preestablished NCI-N87 tumor xenografts were treated with vehicle, trastuzumab (intraperitoneal, 10 mg/kg, 2×/wk × 3), TFBG (subcutaneous, 2.5 mg/kg or intraperitoneal, 50 mg/kg, 1×/day × 21), or a combination of the agents. At the end of experiments, animals were sacrificed and subcutaneous tumors were harvested. Tumor proliferation was assessed by Ki-67 immunohistochemical staining.

**Protein–protein docking**. The binding surface of the PLK1 PBD domain (367–603 aa) and SHCBP1 355–562 aa domain was predicated by the Dock Proteins (ZDOCK)[25] protocol. Briefly, we employed the I-TASSER server[27] to predict the 3D structure of SHCBP1 355–562 aa domain, and then the SHCBP1 355–562 aa domain and PLK1 PBD domain (PBD ID code 1UMW) were respectively defined as receptor and ligand proteins, and no amino acid was predefined as interface residue or paired interacting residue. The docking model with highest ZDOCK score was selected for further experimental validation.

**Virtual screening of inhibitor targeting SHCBP1–PLK1 interaction**. The inhibitors targeting SHCBP1–PLK1 interaction were initially identified by VS using MOE-DOCK Software. Firstly, we identified an inhibitor binding pocket (pocket 5: Val411, Ser412, Lys413, Trp414, Leu490, Leu491, Ala493, Gln531, Asn533, Phe534, Phe535, His538, Lys540, and Arg557) on the binding surface of SHCBP1–PLK1 complex, using MOE-Site Finder plug-ins. Then the bindings of PLK1 pocket 5 with 17,676 small molecules were analyzed using MOE-DOCK and 84 molecules (ΔG > −8.6 kcal/mol) were selected. The properties of the selected molecules, including aqueous solubility (logS), octanol/water partition coefficient (logP), human intestinal absorption (HIA), and CYP2D6 drug interaction, were analyzed using Stardrop software (Version 6.5.0). Finally, 40 compounds were identified for further SPR screening.

**Surface plasmon resonance screening**. The SPR screening of inhibitors targeting SHCBP1–PLK1 was conducted using Biacore T200 system (GE Healthcare). All experiments were carried out with HBS-EP (10 mM HEPES pH 7.4, 150 mM NaCl,

3.4 mM EDTA, and 0.005% surfactant P20) as running buffer with a constant flow rate of 10 μL/min at 25 °C. PLK1, which was diluted with 10 mmol/L sodium acetate buffer (pH 5.0) to a final concentration of 10 μM, was immobilized on a CM5 Sensor Chip surface via covalent linkage to the N terminus of PLK1. Bindings of 40 compounds with PLK1 were performed by passing the molecules (100 μM) through the immobilized PLK1 at the flow rate of 10 μL/min. The association and dissociation time were 120 s, respectively. The compounds which response units >16 RU were screened and the kinetic analyses were performed. For kinetic analyses of the TFBG–PLK1 binding, TFBG was dissolved in the running buffer at different concentrations ranging from 0.78 to 12.5 μM and the kinetic analyses were performed based on the steady-state affinity fit model, according to the procedures described in the software manual.

**Microscale thermophoresis assays**. MST experiments were conducted on a Monolith NT.115 system (NanoTemper Technologies GmbH, Germany), which were used to quantify the interaction between PLK1 and TFBG. PLK1 was labeled with the manufacturer's labeling kits (Monolith$^{TM}$ His-Tag Labeling Kit, Nano-Temper Technologies GmbH, Germany). The PLK1 solutions were prepared in 10 mM PBS (pH 7.4), and TFBG solutions were prepared in 10 mM PBS (5% v/v DMSO, pH 7.4). The concentration of PLK1 is 50 nM, TFBG is titrated from 1.5 × 10$^{-10}$ to 5.0 × 10$^{-6}$ M, the mixed solution of PLK1 and TFBG containing 0.05% v/v Tween 20. The samples were added to the monolith capillaries (MO L022, NanoTemper Technologies) and subsequently subjected to MST analysis. The dissociation constant was determined using a single-site model to fit the curve.

**Microarray analysis**. Total RNA was extracted from 16 cancer and corresponding noncancerous samples from advanced gastric cancer patients, and quality of RNA samples was determined using NanoDrop 2000 and Agilent Bioanalyzer 2100. Samples with RINs > 7.0 were used and mRNA microarray analyses using Affymetrix GeneChip primeview human (Affymetrix, Santa Clara, CA) were performed, according to manufacturer's instructions. Fluorescent signals were scanned using an Affymetrix GeneChip Scanner 3000.

**Statistical analysis**. Statistical analyses were performed using SPSS version 25.0 and GraphPad Prism version 8.0. All data were analyzed for normality using Kolmogorov–Smirnov normality test or Shapiro–Wilk normality test. In terms of data with normality, two-sided Student's $t$ test was used for two groups and one-way ANOVA analysis was performed for multiple groups, followed by the post hoc LSD method (homogeneity of variance) or Tamhane method (heterogeneity of variance). For non-normally distributed values, nonparametric tests were applied. Spearman correlation analysis was used to examine the relativity of gene expression between HER2 and SHCBP1. Kaplan–Meier analysis and log-rank (Mantel–Cox) test were progressed for survival data. Univariate and multivariate analyses were performed using Cox proportional hazards regression models. Statistical significance was considered at $p$ values <0.05.

**Reporting summary**. Further information on research design is available in the Nature Research Reporting Summary linked to this article.

## Data availability

The expression profile microarray data of gastric cancer tissues has been deposited and is available in the ArrayExpress database under accession number E-MTAB-9990. The genomic data for Figs. 1b and 2d are publicly available from GSE66229, GSE54129, GSE62254, GSE15459, and GSE34942. The correlation of SHCBP1 expression with survival of HER2-positive or -negative patients in Supplementary Fig. 2a was analyzed on the website [http://kmplot.com/analysis/]. All relevant data supporting our findings are available within the article and its Supplementary Information files or from the corresponding authors on reasonable request. A reporting summary for this article is available as a Supplementary Information file. Source data are provided with this paper.

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

## Acknowledgements

The authors thank all members in Cuiying Biomedical Research Center, the Department of General Surgery, and the Second Clinical Medical College. We thank Sanduo Zheng for technical support, Yuanxian Xiong and Qi Wang for bioinformatics analysis, and Xiaoying Guan for pathology analysis. This work was supported by National Natural Sciences Foundation of China (31670847); Talent Introduction Plan of the Lanzhou University Second Hospital (yjrckyqdj-2019-01); Gansu Provincial Key Research and Development Program (17YF1FA127); Cuiying Scientific and Technological Innovation Program of Lanzhou University Second Hospital (CY2018-QN01; CY2017-ZD03); Fundamental Research Funds for the Central Universities (lzujbky-2017-78, lzujbky-2020-ct04); and Scientific Research Projects of Colleges in Gansu Province (2019B-013).

## Author contributions

Z.J., H.S., and W.S. conceived the study. Y.L. and G.Z. performed bioinformatics analysis. G.Z., M.L., Z.Y., J.Q., and H.Z. involved in human gastric cancer clinical sample collection and clinical data acquisition. W.S., L.L., and G.Z. performed the FRET and GST pulldown. W.S. and G.Z. performed the yeast two-hybrid and in vitro phosphorylation assays. W.S., Z.M., and G.Z. performed the immunofluorescence, co-immunoprecipitation, immunoblotting, and drug screening. W.S., Z.M., L.L., and H.Z. designed and performed mouse experiments. J.Q., Z.Y., and H.Y. performed the immunohistochemical staining. L.Q. and Z.M. involved in the time-lapse, flow cytometry, and RT-PCR. L.Z. performed the SPR and MST. W.S., Z.M., and M.L. performed the plasmid construction and lentiviral production. X.Z. and X.J. performed data analyses. Z.J., W.S., and G.Z. contributed to the writing and editing of the manuscript.

## Competing interests

The authors declare no competing interests.
