## [Peer Review File · Nature Communications]

Reviewers' Comments:

Reviewer #1:

Remarks to the Author:

In this manuscript (Hyperactivation of HER2-SHCBP1-PLK1 Axis Promotes Tumor Cell Mitosis and Impairs Trastuzumab Sensitivity to Gastric Cancer), Shi et al. provide evidence for an EGF stimulated (HER2 dependent) dissociation of SHCBP1 from SHC1 that triggers phosphorylation of SHCBP1 on Ser-273 and nuclear translocation. They found SHCBP1 localization in mitotic cells was similar to PLK1 (spindle poles, midzone and midbody). LC MS/MS and Two-Hybrid analysis indicated SHCBP1 interacts with PLK1 to stimulate phosphorylation of MISP by PLK1 and promote cell proliferation. The authors found the polo-box domain in PLK1 being required for association with SHCBP1 and used a virtual screen to discover TF3 (a natural compound) is able to disrupt this interaction. Finally, they report TF3 synergizes with trastuzumab in HER2 expressing gastric cancer cells in vitro and in xenograft mouse models.

Although some of the findings are not entirely novel (i.e. EGF triggers dissociation of SHC1 from SHCBP1 was reported in Zheng Y. et al., 2013 Jul 11;499(7457):166-71 and EGF-induced nuclear translocation of SHCBP1 was reported in Liu L. et al., Oncogene 2019 Jan;38(5):747-764), this is an impressive body of work that defines a novel signaling pathway EGF/HER2-SHC1-SHCBP1-PLK1-MISP in HER2 dependent gastric cancer. The results described in this manuscript are of critical importance to our understanding of the basic mechanisms that regulate trastuzumab sensitivity (or SHCBP1 mediated trastuzumab resistance) in HER2 dependent gastric cancer and for translational efforts to improve the efficacy of trastuzumab. The main conclusions are firmly supported by sufficient data of high quality. Finally, this study provides evidence for efficient targeting of HER2 dependent gastric cancer with the trastuzumab/TF3 combination, which makes this a translationally relevant study that is likely to become the basis for a future clinical trial.

Addressing the following points should further strengthen the manuscript:

The role of SHCBP1 in mitosis is not clearly defined. If SHCBP1 is a key regulator of PLK1 activity in the context of HER2 dependent gastric cancer, then knockdown or inhibition of SHCBP1 should result in phenotypes similar to PLK1 inhibition (G2 arrest, spindle checkpoint failure, mitotic catastrophe, etc).

Often the figure legends do not sufficiently describe the experiment. The authors should provide essential experimental information in the figure legends. (See comment below concerning Fig. 5a).

Fig. 1c: it is not clear why the authors focused exclusively on HER2 and not EGFR/HER1, especially because EGFR also came up in Fig.1a. Additionally, Liu L. et al., Oncogene 2019 Jan;38(5):747-764 provide evidence for EGFR stimulating SHCBP1 nuclear translocation. Can similar results be observed when using an EGFR inhibitor (or alternatively neutralizing antibody such as cetuximab)?

Fig. 4c/f and Extended Fig.4b/c: It is not clear how SHCBP1 expression was quantified. Please elaborate on the criteria used to quantify. y-axis label is misleading. Better: "SHCBP1 positivity of SNU-216 cells (%)". Fig. 4e: The label "WT" is misleading because a wild-type is not a control for trastuzumab. A negative control for Trast should be either vehicle control or unrelated antibody. Fig. 4g: It seems the blots are flipped because the lamin B signal is in the Cyto and the GAPDH signal in the Nuc fraction.

Fig. 5a: If these cells were synchronized in M phase then this should be indicated in the figure legend.

Fig. 5g: Need to show efficiency of SHCBP1 knockdown.

Fig. 5j: figure legend (page 34, line 886) is misleading. The notion of the FLAG antibody detecting MISP phosphorylation is not supported. Should be reworded to: "Detection of MISP mobility shift...".

Fig. 7g: the DAPI staining in the Trast treatment groups looks a bit odd because nuclei cores seem

to be void of DAPI staining. Also, the staining in blue and green looks more diffuse as compared to the Ctrl group. Please provide better images with consistent nuclear staining.
Fig. 7h: it is not clear what criteria were used to define "nuclear SHCBP1" Based on the images in Fig. 7g, there is no nuclear DAPI in Trast treated cells either. This issue needs to be resolved.

Mass Spectrometry Analysis (Math & Meth, page 24, line 617f): Lysis buffer containing 1M Tris, 1M NaCl, 0.5M EDTA and 10% Triton has extremely high stringency and therefore is not suited for investigating interactions. Please clarify.

Reviewer #2:
Remarks to the Author:
Review of paper by Shi W et al

Key results: Please summarize what you consider to be the outstanding features of the work. The article explores the mechanistic of the downstream pathway of SHCBP1 and discovers its role in resistance to trastuzumab therapy in HER2 positive gastric cancers. The authors identify and test a potential therapeutic, TF3, to counteract the resistance mechanism which has shown effect in xenograft mouse models.

Validity: Does the manuscript have flaws which should prohibit its publication? If so, please provide details.
The manuscript is scientifically sound and does not have major flaws prohibiting publication. However, the manuscript needs revisions before publication can be recommended.

Originality and significance and referencing of previous literature
In the discussion, please comment on results by other researchers: SHCBP1 correlated to decreased overall survival in other cancer types such as NSCLC and breast cancer (1). In breast cancer, correlation with HER2 expression has also been established (2). Taking both studies together it has been shown prior to this study that SHCBP1 translocates to the nucleus and affects cell mitosis, albeit through other pathways.
SHCBP1 has also been identified to play a role in cisplatin induced apoptosis resistance in NSCLC (3).
SHCBP-1 has earlier been identified in other gastric cancer cell lines (4). In another form of upper gastrointestinal cancer, esophageal squamous cell cancer (ESCC), SHCBP1 has also found to be correlated to clinical variables and in xenograft models SHCBP1 depletion by shRNA lentivirus has also shown to affect ESCC tumor growth in mice (5).

Subjective opinion:
While some conclusions in the article to some degree have been described before this article still has marked novelty due to:
1. The use of several methodologies to in-depth elucidate SHCBP1 function including molecular interaction studies, mechanistic studies, studies in gastric cancer cell lines and in-vivo model systems and small clinical cohorts as well as drug discovery.
2. Most importantly establishing concrete evidence and rationale for a population of patients in which SHCBP1 targeting is relevant.
3. Discovering, examining and suggesting relevant therapy.

While the study examines gastric cancer it also has important relevance to other HER2 positive cancers where HER2 targeted therapy is used such as breast cancer, in which SHCBP1 and HER2 expression also correlate(2).

Introduction

(1) In the introduction the authors state "Trastuzumab is the only effective anti-HER2 therapy used in gastric cancer clinical treatment, and all the other HER2-targeted molecules, including lapatinib, pertuzumab and trastuzumab emtansine (T-DM1), have failed to show a survival benefit" (Page 3, lines 52ff). Please update and cite study showing survival benefit with trastuzumab deruxtecan in HER2 pos recurrent gastric cancer

(2) In page 3, lines 55ff, authors state: "Even for trastuzumab, the overall response rate is reportedly only 47%, and the median overall survival is improved by 2.7-month in patients with trastuzumab plus chemotherapy compared to those with chemotherapy alone 3,8, which is significantly shorter than the 4.8-month improvement in breast cancer patients with standard chemotherapy plus trastuzumab."

Not valid to compare across studies in general and in particular to compare median survivals which are merely survival at an arbitrary selected single timepoint. HR reflects better the survival over time in a given population. In fact, in the mBC trial (Slamon) HR was 0.80 and in the gastric/GEJ (Bang) HR was 0.74. Thus, you could equally argue that trastuzumab is more active in gastric/GEJ. Please omit this paragraph comparing efficacy in breast and gastric cancer or modify with other data backing the statement.

(3) In page 3, lines 60ff, it is stated that "The mechanisms underlying insufficient sensitivity of HER2-directed therapy to gastric cancer are proposed to be aberrant activation of HER2 and downstream signaling, including amplification, upregulation or mutation of HER2, KRAS, PIK3CA, AKT and PTEN, which make it difficult to inhibit activation of downstream signaling and cell growth using trastuzumab only". Comment: Similar mechanisms might be operating in breast cancer. Please provide persuading arguments that these mechanisms are specific for gastric cancer and not mechanisms operating in general for resistance to trastuzumab or omit.

(4) Page 3, lines 67: "However, no positive data have been obtained to date, and none of the new anti-HER2 treatment strategies improved survival significantly". Comment: As above, trastuzumab deruxtecan resulted a survival benefit in recurrent HER positive gastric/GEJ adenocarcinoma. Please update and cite the above-mentioned paper.

Data & methodology:

(1) Please comment on the cut-off values for high and low SHCBP1 expression used in figure 2 and how this value was derived/chosen and consider adding the definition in the main text and not only in supplementary figure 2b.

(2) In the methods section of "Surface Plasmon Resonance Screening" you mention that you chose a cut-off value of >16 RU. Looking at supplementary extended data fig. 7e, eight compounds seem to pass this threshold. Please elaborate on how and why TF3 was chosen from these eight compounds, was it based on Kd value or something else?

(3) Regarding the inhibitory effect of TF3 (page 13, lines 326ff). IC50 of an inhibitor of cell proliferation in the micromolar range should not be categorized as a potent inhibitor but rather moderate inhibitor of PLK1-SHCBP1 interaction. Drugs reaching the clinic typically have IC50 in the nmolar range.

Appropriate use of statistics and treatment of uncertainties:

Please include in the statistics section the methods used for determining if the prerequisite assumptions for conducting one-way ANOVA were fulfilled: (1) That the dependent variable is approximately normal distributed for each categorical variable.

(2) That there is homogeneity of variances.

Spearman coefficient of 0.4 is normally regarded as a weak to moderate correlation – please indicated that the correlation between HER2 and SHCBP1 expression is weak-to-moderate which is evident from the quite scattered data points in the correlation plot (page 6, line 132, figure 2f)

The hypotheses that high SHCBP1 is associated with decreased survival in HER pos gastric cancer and that high SHCBP1 confers resistance to trastuzumab are biased by the small number of patients, no multivariate analysis, no validation in an independent patient cohort, no prespecified cut off point for high vs low SHCBP1 (page 6, line 137ff). These limitations need to be acknowledged. Also – as mentioned above – comment on data indicating that SHCBP1 correlates to decreased overall survival in other cancer types such as NSCLC and breast cancer (1,2)

Conclusions: Do you find that the conclusions and data interpretation are robust, valid and reliable?

The conclusions are robust, but caution and conservatism regarding interpreting p values of one-way ANOVA if prerequisites are not fulfilled should be exhibited.

Suggested improvements:

The reviewer has no suggestions for additional experiments. But for transparency of the methodology in this work please take the points raised in "Data & methodology:" into consideration as well as the statistical points. Further comments and discussion on the previous literature in context of the results in this manuscript should be done.

Clarity and context: The abstract is clear and to the point. The introduction needs modification as indicated above. The discussion is also clear, but do not provide enough context of previous knowledge in regards of SHCBP1 and cancer. List of references have been cut in two and should be merged properly.

Please indicate any particular part of the manuscript, data, or analyses that you feel is outside the scope of your expertise, or that you were unable to assess fully.

Affinity Purification and Mass Spectrometry Analysis, Protein-Protein Docking, Virtual Screening of Inhibitor Interaction and Surface Plasmon Resonance Screening are analyses outside my expertise which I am able to assess fully.

References

1. Liu L, Yang Y, Liu S, et al. EGF-induced nuclear localization of SHCBP1 activates β -catenin signaling and promotes cancer progression. *Oncogene*. 2019;38(5):747-764. doi:10.1038/s41388-018-0473-z
2. Feng W, Li H, Xu K, et al. SHCBP1 is over-expressed in breast cancer and is important in the proliferation and apoptosis of the human malignant breast cancer cell line. *Gene*. 2016;587(1):91-97. doi:10.1016/j.gene.2016.04.046
3. Zou A, Wu A, Luo M, Zhou C, Lu Y, Yu X. SHCBP1 promotes cisplatin induced apoptosis resistance, migration and invasion through activating Wnt pathway. *Life Sci*. 2019;235:116798. doi:10.1016/j.lfs.2019.116798
4. Dong YD, Yuan YL, Yu HB, Tian GJ, Li DY. SHCBP1 is a novel target and exhibits tumor-promoting effects in gastric cancer. *Oncol Rep*. 2019;41(3):1649-1657. doi:10.3892/or.2018.6952
5. Ren C, Zhou Z, Wang X, Hua X, Zou M, Zhang X. SHCBP1 Promotes the Progression of Esophageal Squamous Cell Carcinoma Via the TGF β Pathway. *Appl Immunohistochem Mol Morphol*. 2020; Publish Ahead of Print(00):1-8. doi:10.1097/pai.0000000000000858
6. Shitara K, Bang Y-J, Iwasa S, et al. Trastuzumab Deruxtecan in Previously Treated HER2-Positive Gastric Cancer *N Engl J Med* 2020;382:2419-30. DOI: 10.1056/NEJMoa2004413

Reviewer #3:

Remarks to the Author:

The presented manuscript by Shi et al. investigates the function of SHCBP1 in HER2-amplified gastric cancers. The authors proposed a model by which SHC1 brings in a scaffold protein SHCBP1 to interact with HER2. Interestingly, SHCBP1 (BP1) translocates to the nucleus to promote cell cycle progression via its interaction with PLK1.

Several critical mechanistic issues are noted and need close attention for the authors to clarify:

1) Line 90-97 explains how the authors identified BP1: the bioinformatics process of zeroing in on BP1 is somewhat arbitrary and circumspect. There is no direct evidence in the gene expression data that BP1 is of any importance for HER2+ GC. The authors need to explain better the logic of investigating SHC1 binding partners.

2) Line 96: the term amplified is being used incorrectly.

3) Gene expression in Fig. 1c is not same as interactions between proteins- not convincing. Of all data shown, only 1D is somewhat convincing. How other components of HER2 signaling complex correlate with HER2 mRNA. Clarity is needed whether the authors are showing proteins or mRNA vs. copy number.

- 4) There is no explanation as to why EGF was used: there is no change in IP of SHCBP1 with or without EGF. Plus, EGF does not bind to HER2.
- 5) It is unclear if the interaction of SHC1 and its BP1 is a new knowledge obtained in this paper, no citation is provided.
- 6) Line 122: how unique is the role of SHCBP1 as a HER2 effector needs to be spelled out. It seems a gross overstatement
- 7) Fig. 2: the term "paracancer" is confusing and should be replaced with "adjacent normal" if this is what the authors implied.
- 8) Up until Fig. 4, the evidence is guiding to a conclusion that SHCBP1 is an important modulator of HER2 function, thus regulating sensitivity to trastuzumab. However, starting from Fig4, the story completely abandons the idea of HER2 function and goes on to describe SHCBP1 nuclear importation. Proper signaling studies and evaluation of HER2 trafficking and stability in the presence or absence of SHCBP1 are needed to fully exclude the effects on the trastuzumab main target.
- 9) The interpretation of the S273A mutation is not as straightforward as the authors imply: the abrogation of this phosphosite could easily affect its interaction with SHC1/HER2, cause protein misfolding, and not only nuclear translocation. Detailed biochemistry of this mutant is needed to draw this conclusion. In contrast, a more plausible theory could be in place such as stabilization of HER2 or its more robust recycling to the plasma membrane due to its interaction with SHC/SHCBP1. In fact, FRET data (fig.3) shows predominantly membrane colocalization of HER and SHCBP1, nothing is present to appreciable levels in the nucleus.
- 10) Functional domains of SHCBP1 are not characterized. No evidence for nuclear localization signal is shown.
- 11) The abrogation of nuclear fraction of SHCBP1 in cells treated with trastuzumab (Fig. 4E) further argues that BP1 is intimately attached to HER2 signaling complex and is removed from the nucleus by HER2 crosslinking. In these experiments, again, the stimulation of cells with EGF does not provide mechanistic explanation as to why it would change BP1 redistribution between cyto and nuc compartments.
- 12) The biochemistry of nuclear interactors with SHCBP1 is interesting although not convincing the mechanism of TF3 is necessarily related to its action on nuclear proteins. Again, in line with the comments on mechanism above, trastuzumab and TF3 combo produces remarkable HER2 depletion (Fig.6), which is a likely mechanism for the synergy. Testing of TF3 in disrupting SHC1/SHCBP1/HER2 complexes will clarify and support this more interesting and signaling-specific mechanism. Additional detailed signaling and HER2 trafficking characterization is needed for the TF3 mechanism.

Responses to Referees Letter
(NCOMMS-20-38715A)

Thank you and reviewers very much for your kind and valuable comments and suggestions to our manuscript (NCOMMS-20-38715A). They are truly helpful to improve the quality of our work. We have revised the manuscript according to these suggestions and comments. Based on the suggestions from Reviewer #1 and #3, we performed additional experiments to strengthen the mechanism of HER2-SHCBP1-PLK1 axis. According to the comments of Reviewer #2, we corrected the inaccurate description of our findings, especially the clinical relevance of SHCBP1 and HER2-positive patients. Itemized point-to-point changes and responses are listed below with comments in black and responses in blue. All changed or added text in the manuscript are quoted as red in a revised manuscript uploaded as a file named 'Marked revised manuscript.docx'.

.....

Responses to reviewer comments:

Reviewer #1:

Although some of the findings are not entirely novel (i.e. EGF triggers dissociation of SHC1 from SHCBP1 was reported in Zheng Y. et al., 2013 Jul 11;499(7457):166-71 and EGF-induced nuclear translocation of SHCBP1 was reported in Liu L. et al., Oncogene 2019 Jan;38(5):747-764), this is an impressive body of work that defines a novel signaling pathway EGF/HER2-SHC1-SHCBP1-PLK1-MISP in HER2 dependent gastric cancer. The results described in this manuscript are of critical importance to our understanding of the basic mechanisms that regulate trastuzumab sensitivity (or SHCBP1 mediated trastuzumab resistance) in HER2 dependent gastric cancer and for translational efforts to improve the efficacy of trastuzumab. The main conclusions are firmly supported by sufficient data of high quality. Finally, this study provides evidence for efficient targeting of HER2 dependent gastric cancer with the trastuzumab/TF3 combination, which makes this a translationally relevant study that is likely to become the basis for a future clinical trial.

Addressing the following points should further strengthen the manuscript:

1. The role of SHCBP1 in mitosis is not clearly defined. If SHCBP1 is a key regulator of PLK1 activity

in the context of HER2 dependent gastric cancer, then knockdown or inhibition of SHCBP1 should result in phenotypes similar to PLK1 inhibition (G2 arrest, spindle checkpoint failure, mitotic catastrophe, etc).

RESPONSE: Thanks for your comments and suggestions! We have performed additional experiments to validate the regulatory function of SHCBP1 in mitosis of gastric cancer cells according to the comments. Flow cytometry detection of cell cycle indicated that the knockdown of SHCBP1 induced a G2/M arrest of cancer cells (which shown in supplementary Fig. 5b, c). We monitored the division of SHCBP1 knockdown cells by time-lapse microscopy and observed a delay in mitotic progression (Supplementary Fig. 5d). Finally, we also detected the mitotic spindle stained for α -tubulin and the centrosomal marker γ -tubulin in SHCBP1 depleted cells, which revealed a marked increase of the cells with defective spindle or multipolar spindle (Supplementary Fig. 5e, f). All above new findings strongly supported that SHCBP1 is a key regulator of mitotic progression in gastric cancer cells. We have added the description of the new findings in the **Results** part of the manuscript.

The added description as follow:

“Previous studies demonstrated that SHCBP1 regulates cytokinesis completion by the interaction with centralspindlin complex composed of MKLP1 and MgcRacGAP^{22,23}, but the role of SHCBP1 in metaphase is still unknown. To further investigate the mechanisms of how SHCBP1 contributes to cell division in metaphase, we knocked down the SHCBP1 expression in SNU-216 cells and detected the changes of cell mitotic progression. Cell cycle assay using flow cytometry showed that SHCBP1 depletion induced a significant G2/M arrest (Supplementary Fig. 5b, c). We monitored the division of SHCBP1 knockdown cells by time-lapse microscopy and observed an obvious delay in mitotic progression (Supplementary Fig. 5d). Furthermore, close inspection of the mitotic spindle stained with α -tubulin and the centrosomal marker γ -tubulin showed a marked increase of defective spindle and multipolar spindle in SHCBP1 knockdown cells (Supplementary Fig. 5e, f). These results suggested that SHCBP1 is essential for proper spindle formation in mitotic progression during metaphase.”

2. Fig. 1c: it is not clear why the authors focused exclusively on HER2 and not EGFR/HER1, especially because EGFR also came up in Fig.1a. Additionally, Liu L. et al., *Oncogene* 2019 Jan;38(5):747-764 provide evidence for EGFR stimulating SHCBP1 nuclear translocation. Can similar

results be observed when using an EGFR inhibitor (or alternatively neutralizing antibody such as cetuximab)?

RESPONSE: Thanks for your suggestion! We agreed that EGFR/HER1 activation may also induce the SHCBP1 nuclear translocation and activation of downstream PLK1-MISP signaling. We have performed additional experiments to validate it. As shown in the **Fig. 1 for reviewers** below, we found that EGFR inhibitor (EGFR-IN-12) significantly blocked EGF-induced SHCBP1 nuclear translocation and MISP phosphorylation, demonstrating EGFR dependence for SHCBP1/PLK1/MISP axis activation. But it must be emphasized that HER2-directed therapy trastuzumab is approved as first-line treatment for patients with metastatic gastric cancer, while the EGFR-targeted therapies have failed to show a survival benefit in gastric cancer to date (Lordick et al., 2013, Satoh et al., 2015). Thus, we focused on HER2 instead of EGFR to investigate the role of SHCBP1 in drug resistance and sought to identify SHCBP1 inhibitors combined with trastuzumab to develop new treatment strategies for gastric cancer.

References:

Lordick F, Kang YK, Chung HC, et al. Capecitabine and cisplatin with or without cetuximab for patients with previously untreated advanced gastric cancer (EXPAND): a randomised, open-label phase 3 trial. *Lancet Oncol.* 2013;14(6):490-499.

Satoh T, Lee KH, Rha SY, et al. Randomized phase II trial of nimotuzumab plus irinotecan versus irinotecan alone as second-line therapy for patients with advanced gastric cancer. *Gastric Cancer.* 2015;18(4):824-832.

Fig. 1 for reviewers. The effects of EGFR inhibitor (EGFR-IN-12) on SHCBP1 nuclear translocation and MISP phosphorylation. (A) Immunoblotting analysis of SHCBP1 in nuclear (Nuc) and cytosol extracts (Cyto) of SNU-216 cells treated with 100 ng/mL EGF and/or 10 μM EGFR inhibitor. (B) Immunoblotting assays of MISP phosphorylation in SNU-216 cells following EGF and/or EGFR inhibitor treatment.

3. Fig. 4c/f and Extended Fig.4b/c: It is not clear how SHCBP1 expression was quantified. Please elaborate on the criteria used to quantify. y-axis label is misleading. Better: “SHCBP1 positivity of SNU-216 cells (%)”.

RESPONSE: Sorry for our confused description and thanks for your suggestion! The SHCBP1 nuclear translocation was detected using SHCBP1 immunofluorescence and the positivity of SHCBP1 in the nucleus was quantified using Image-Pro Plus software. The mean fluorescence intensity (MFI) of SHCBP1 in the nucleus was quantified and the MFI > 10 was identified as positive cells. We have added the details in the *Methods* part of the manuscript, and we also changed the y-axis label of Fig. 4c/f and supplementary Fig. 4b/c as your suggestion.

The added description as follow:

“The positivity of SHCBP1 in the nucleus was detected by SHCBP1 immunofluorescence and was quantified using Image-Pro Plus software. The mean fluorescence intensity (MFI) of SHCBP1 in the nucleus was quantified and the MFI > 10 was identified as the positive cells.”

4. Fig. 4e: The label “WT” is misleading because a wild-type is not a control for trastuzumab. A negative control for Trast should be either vehicle control or unrelated antibody.

RESPONSE: Thanks a lot for your suggestions! We have corrected the misleading label in Fig. 4e.

5. Fig. 4g: It seems the blots are flipped because the lamin B signal is in the Cyto and the GAPDH signal in the Nuc fraction.

RESPONSE: Sorry for our mistake, the misleading label of the blots have been corrected in Fig. 4g.

6. Fig. 5a: If these cells were synchronized in M phase then this should be indicated in the figure legend.

RESPONSE: Thanks for your suggestion! We have indicated the treatment of M phase synchronization in the figure legend.

The figure legend of Fig. 5a was added description as follow:

“Co-immunoprecipitation assays of Flag/HA-tagged SHCBP1, PLK1 or MISP co-expressed in

HEK293T cells which were synchronized to the mitotic phase using 40 ng/mL nocodazole (NOC).”

7. Fig. 5g: Need to show efficiency of SHCBP1 knockdown.

RESPONSE: We have added the western blotting results indicating the efficiency of SHCBP1 knockdown in Fig. 5g.

8. Fig. 5j: figure legend (page 34, line 886) is misleading. The notion of the FLAG antibody detecting MISP phosphorylation is not supported. Should be reworded to: “Detection of MISP mobility shift...”.

RESPONSE: Thanks a lot! The confusing descriptions have been corrected.

The figure legend was corrected as follow:

“Detection of MISP mobility shift in cells transfected with Flag-tagged MISP (WT) or mutant MISP (3A: S394A, S395A and S397A) by immunoblotting using Flag antibody.”

9. Fig. 7g: the DAPI staining in the Trast treatment groups looks a bit odd because nuclei cores seem to be void of DAPI staining. Also, the staining in blue and green looks more diffuse as compared to the Ctrl group. Please provide better images with consistent nuclear staining.

RESPONSE: Thanks for your suggestion! We have replaced the images of Fig. 7g with better images.

10. Fig. 7h: it is not clear what criteria were used to define “nuclear SHCBP1” Based on the images in Fig. 7g, there is no nuclear DAPI in Trast treated cells either. This issue needs to be resolved.

RESPONSE: Thanks a lot for your suggestion! We have replaced Fig. 7g with better images with clear DAPI-labeled nucleus. The nuclear SHCBP1 was quantified using Image-Pro Plus software. The fluorescence intensity of SHCBP1 in the nucleus was quantified and the nuclear fluorescence intensity > 10 was identified as positive cells. The details were added in the *Figure legends* and *Methods* in the text of the manuscript.

The revised figure legend as follow:

“Tissues were co-stained with SHCBP1 antibody (green), HER2 antibody (red), and DAPI (blue). The positivity of SHCBP1 in the nucleus was quantified using Image-Pro Plus software.”

11. Mass Spectrometry Analysis (Math & Meth, page 24, line 617f): Lysis buffer containing 1M Tris, 1M NaCl, 0.5M EDTA and 10% Triton has extremely high stringency and therefore is not suited for investigating interactions. Please clarify.

RESPONSE: We apologize for our mistake. We have misworded the concentration of storage solution as final concentration of the lysis buffer. The correct concentration is 50 mM Tris-HCl (pH = 7.4), 150 mM NaCl, 1 mM EDTA, 0.2% TritonX-100. We have corrected these in the *Methods* of the Manuscript.

The revised description as follow:

“Cells were lysed with lysis buffer [50 mM Tris-HCl (pH = 7.4), 150 mM NaCl, 1 mM EDTA, 0.2% TritonX-100] containing protease inhibitor.”

Reviewer #2:

1. In the discussion, please comment on results by other researchers: SHCBP1 correlated to decreased overall survival in other cancer types such as NSCLC and breast cancer (1). In breast cancer, correlation with HER2 expression has also been established (2). Taking both studies together it has been shown prior to this study that SHCBP1 translocates to the nucleus and affects cell mitosis, albeit through other pathways. SHCBP1 has also been identified to play a role in cisplatin induced apoptosis resistance in NSCLC (3). SHCBP-1 has earlier been identified in other gastric cancer cell lines (4). In another form of upper gastrointestinal cancer, esophageal squamous cell cancer (ESCC), SHCBP1 has also found to be correlated to clinical variables and in xenograft models SHCBP1 depletion by shRNA lentivirus has also shown to affect ESCC tumor growth in mice (5).

RESPONSE: Thanks for your comments! We have added further discussions in the *Discussion* part according to your suggestion and added relevant references (Reference 22, 23, 31-35).

The added discussion as follow:

“Previous studies also demonstrated the oncogenic role of SHCBP1 in non-small cell lung carcinoma (NSCLC) and breast cancer. It was found that SHCBP1 was upregulated and correlated with poorer survival of patients with NSCLC and breast cancer^{31,32}. Moreover, SHCBP1 was found strongly associated with the HER2 expression in breast cancer³². In here, we demonstrated a positive correlation between HER2 and SHCBP1 expression in gastric cancer, which suggested that the SHCBP1 may be

involved in HER2 promoted tumorigenesis. We knocked down the SHCBP1 expression and found that SHCBP1 depletion inhibited the cell proliferation and sensitized the cell to trastuzumab in gastric cancer cell SNU-216 and NCI-N87. Consistently, others also reported that SHCBP1 knockdown inhibited the proliferation and metastasis of gastric cancer cell MGC-803 and SGC-7901, and suppressed the proliferation and motility of esophageal squamous cell carcinoma (ESCC) cells^{33,34}. These findings demonstrated the oncogenic role of SHCBP1 in a variety of human cancers.”

“Previously, SHCBP1 was identified to play a role in cisplatin induced apoptosis resistance in NSCLC, indicating the regulatory function of SHCBP1 in drug resistance to cancers ³⁵.”

“Interestingly, previous observations also indicated that EGF stimulation can cause SHCBP1 redistribution to the nucleus. SHCBP1 interacts with β -catenin for nuclear translocation to activate β -catenin signaling³¹”.

“Indeed, SHCBP1 was reported be responsible for cell mitosis during cytokinesis, which was colocalized at the midbody with MgcRacGAP and MKLP1, where it forms centralspindlin and promotes the ingression of the cytokinetic furrow^{22,23}.”

2. In the introduction the authors state "Trastuzumab is the only effective anti-HER2 therapy used in gastric cancer clinical treatment, and all the other HER2-targeted molecules, including lapatinib, pertuzumab and trastuzumab emtansine (T-DM1), have failed to show a survival benefit" (Page 3, lines 52ff). Please update and cite study showing survival benefit with trastuzumab deruxtecan in HER2 pos recurrent gastric cancer⁶.

RESPONSE: Thanks a lot for your suggestion! It has been added in the introduction as “Trastuzumab and trastuzumab deruxtecan are effective anti-HER2 therapy showing survival benefit in gastric cancer ⁴⁻⁶.” and the references were also cited (Reference 6).

3. In page 3, lines 55ff, authors state: "Even for trastuzumab, the overall response rate is reportedly only 47%, and the median overall survival is improved by 2.7-month in patients with trastuzumab plus chemotherapy compared to those with chemotherapy alone 3,8, which is significantly shorter than the 4.8-month improvement in breast cancer patients with standard chemotherapy plus trastuzumab." Not valid to compare across studies in general and in particular to compare median survivals which are

merely survival at an arbitrary selected single timepoint. HR reflects better the survival over time in a given population. In fact, in the mBC trial (Slamon) HR was 0.80 and in the gastric/GEJ (Bang) HR was 0.74. Thus, you could equally argue that trastuzumab is more active in gastric/GEJ. Please omit this paragraph comparing efficacy in breast and gastric cancer or modify with other data backing the statement.

RESPONSE: Thanks for your suggestion! We have deleted the inappropriate comparative descriptions of trastuzumab in gastric cancer and breast cancer treatment. We have added a statement of the importance of trastuzumab in gastric cancer treatment.

The added description as follow:

“A lot of HER2-directed therapies have been used in the treatment of HER2-positive cancers. Trastuzumab, pertuzumab, lapatinib and T-MD1 are used in HER2-positive breast cancer³. Trastuzumab and trastuzumab deruxtecan are effective anti-HER2 therapy showing survival benefit in gastric cancer⁴⁻⁶. In the trastuzumab for gastric cancer (ToGA) trial, the overall survival of patients with HER2-positive advanced gastric cancer was improved 2.7 months when trastuzumab was combined with conventional chemotherapy, compared to chemotherapy alone. These results led to the approval trastuzumab, which is now the first-line treatment in combination with oxaliplatin or fluorouracil chemotherapy for patients with HER2-positive metastatic gastric cancer⁴. However, many patients with HER2-positive gastric cancer still succumb to their disease following trastuzumab therapy, one of the main reasons is the intrinsic and secondary resistance.”

4. In page 3, lines 60ff, it is stated that "The mechanisms underlying insufficient sensitivity of HER2-directed therapy to gastric cancer are proposed to be aberrant activation of HER2 and downstream signaling, including amplification, upregulation or mutation of HER2, KRAS, PIK3CA, AKT and PTEN, which make it difficult to inhibit activation of downstream signaling and cell growth using trastuzumab only". Comment: Similar mechanisms might be operating in breast cancer. Please provide persuading arguments that these mechanisms are specific for gastric cancer and not mechanisms operating in general for resistance to trastuzumab or omit.

RESPONSE: Thanks a lot for your suggestion! We have added new mechanisms of trastuzumab resistance which are specific for gastric cancer and the relevant references were cited in the manuscript

(Reference 9-15).

The revised description as follow:

“HER2-positive gastric cancer has been found to share some of these mechanisms, but also manifests specific mechanisms of resistance to trastuzumab. For example, intratumoral HER2 heterogeneity is more frequent in gastric cancer than in breast cancer, with values ranging from 23% to 79%⁹. Besides, loss of HER2 protein expression in pre- and post-treatment of gastric cancer patients is also a main reason of trastuzumab resistance¹⁰. Other proposed resistance mechanisms include alterations in HER2 downstream signaling and bypass pathways, such as upregulation of kallikrein 10 (KLK10), metastasis associated in colon cancer 1 (MACC1), C-Maf-inducing protein (CMIP), hyperactivation of HER4-YAP1 axis, TNF α -induced Mucin 4 (MUC4) overexpression¹¹⁻¹⁵.”

5. Page 3, lines 67: "However, no positive data have been obtained to date, and none of the new anti-HER2 treatment strategies improved survival significantly". Comment: As above, trastuzumab deruxtecan resulted a survival benefit in recurrent HER positive gastric/GEJ adenocarcinoma. Please update and cite the above-mentioned paper.

RESPONSE: Thanks a lot! It has been updated and the relevant reference was cited. (Reference 6).

The updated description as follow:

“Although trastuzumab deruxtecan has been proved resulting a survival benefit in patients with HER2-positive gastric cancer, no other new data have been obtained to date, and none of the new anti-HER2 treatment strategies improved survival significantly enough to justify registration^{6,20}.”

6. Please comment on the cut-off values for high and low SHCBP1 expression used in figure 2 and how this value was derived/chosen and consider adding the definition in the main text and not only in supplementary figure 2b.

RESPONSE: Thanks a lot! We divided H-score < 70 as SHCBP1 low group and H-score \geq 70 as SHCBP1 high group according to the median value of SHCBP1 H-score. This definition has been added in the *Method* and *main text* in the manuscript.

The added description as follow:

“The H-score < 70 is divided as SHCBP1 low expression group and H-score \geq 70 as SHCBP1 high

expression group according to the median value of SHCBP1 H-score.”

7. In the methods section of “Surface Plasmon Resonance Screening” you mention that you chose a cut-off value of >16 RU. Looking at supplementary extended data fig. 7e, eight compounds seem to pass this threshold. Please elaborate on how and why TF3 was chosen from these eight compounds, was it based on K_D value or something else?

RESPONSE: Thanks a lot! Firstly, we screened the inhibitor using SPR at single concentration and 8 compounds which response units > 16 RU were found. Then, we identified 4 compounds according to the binding and dissociation rate of the compounds, and the K_D value were detected. Finally, we screened out TFBG (TF3) as the best inhibitor of SHCBP1-PLK1 due to its minimum K_D value. The K_D values of the 4 compounds are Protodioscin: 2.972 M, Ginsenoside Rg3: 597.4 μ M, A1874: 21.11 μ M and TFBG: 466.6 nM. We have added the description in the **Results** of the manuscript.

The added description as follow:

“Finally, a natural product, theaflavine-3, 3'-digallate (TFBG), which selectively and efficiently targeted PLK1, was identified as the inhibitor according to its minimum K_D value.”

8. Regarding the inhibitory effect of TF3 (page 13, lines 326ff). IC50 of an inhibitor of cell proliferation in the micromolar range should not be categorized as a potent inhibitor but rather moderate inhibitor of PLK1-SHCBP1 interaction. Drugs reaching the clinic typically have IC50 in the nmolar range.

RESPONSE: Thanks a lot! We agree that TFBG (TF3) is a moderate inhibitor of PLK1-SHCBP1 interaction and we have deleted the inappropriate description. It is true that single agent TFBG can inhibit the cancer growth moderately, but TFBG and trastuzumab combination can significantly inhibit the tumor growth and TFBG can sensitize trastuzumab at a lower concentration.

The revised description as follow:

“Taken together, these data demonstrate that TFBG is a selective inhibitor targeting the PLK1-SHCBP1 interaction and displays the potential for use as an anticancer drug.”

9. Please include in the statistics section the methods used for determining if the prerequisite

assumptions for conducting one-way ANOVA were fulfilled: (1) That the dependent variable is approximately normal distributed for each categorical variable. (2) That there is homogeneity of variances.

RESPONSE: Thanks a lot! The revised description of statistical methods was added in the *Method* part of the manuscript.

The add description as follow:

“Statistical analyses were performed using SPSS version 25.0 and GraphPad Prism version 8.0. All data were analyzed for normality using Kolmogorov-Smirnov normality test or Shapiro-Wilk normality test. In terms of data with normality, two-tailed Student's t-test was used for two groups and one-way ANOVA analysis was performed for multiple groups, followed by the post hoc LSD method (homogeneity of variance) or Tamhane method (heterogeneity of variance). For non-normally distributed values, non-parametric tests were applied. Spearman correlation analysis was used to examine the relativity of gene expression between HER2 and SHCBP1. Kaplan-Meier analysis and Log-rank (Mantel-Cox) test were progressed for survival data. Univariate and multivariate analyses were performed using Cox proportional hazards regression models. Statistical significance was considered at p values less than 0.05.”

10. Spearmann coefficient of 0.4 is normally regarded as a weak to moderate correlation – please indicated that the correlation between HER2 and SHCBP1 expression is weak-to-moderate which is evident from the quite scattered data points in the correlation plot (page 6, line 132, figure 2f).

RESPONSE: Thanks for your suggestion! We have corrected the inappropriate description.

The revised description as follow:

“We performed immunohistochemical and immunofluorescent analysis of the gastric cancer TMAs to determine the clinical relevance of SHCBP1 and HER2, demonstrating a weak to moderate correlation between HER2 and SHCBP1 expression.”

11. The hypotheses that high SHCBP1 is associated with decreased survival in HER pos gastric cancer and that high SHCBP1 confers resistance to trastuzumab are biased by the small number of patients, no multivariate analysis, no validation in an independent patient cohort, no prespecified cut off point for

high vs low SHCBP1 (page 6, line 137ff). These limitations need to be acknowledged. Also – as mentioned above – comment on data indicating that SHCBP1 correlates to decreased overall survival in other cancer types such as NSCLC and breast cancer (1,2).

RESPONSE: Thanks a lot! We have furtherly performed the univariate and multivariate analyses of SHCBP1 and patients with HER2 positive gastric cancer, and demonstrated SHCBP1 expression was an independent prognostic factor for HER2-positive gastric cancer patients. The result was added in Supplementary Table 1 and the description was added in the main text. For other limitations of clinical analysis in our paper, we have added a statement in this regard in the *Discussion* part of the manuscript. Please see the added description below.

The cut off point for high vs low SHCBP1 was described both in the *Method* and *Results* of the manuscript. Please review our above response of comments 6.

The comment on data indicating that SHCBP1 correlates to decreased overall survival in other cancer types such as NSCLC and breast cancer was added in the *Discussion*. Please review our above response of comments 1.

The added description as follow:

“Furthermore, the univariate and multivariate analyses of patients from the gastric cancer TMAs demonstrated SHCBP1 expression was an independent prognostic factor for HER2-positive gastric cancer patients (Supplementary Table 1).”

“However, more clinical analysis of correlation between SHCBP1 and HER2-positive patients who received trastuzumab therapy have not been conducted in here due to the shortage of the patients. A large-scale clinical study is required in the future to validate the importance of SHCBP1 in trastuzumab-based therapy.”

Reviewer #3:

1. Line 90-97 explains how the authors identified BP1: the bioinformatics process of zeroing in on BP1 is somewhat arbitrary and circumspect. There is no direct evidence in the gene expression data that BP1 is of any importance for HER2+ GC. The authors need to explain better the logic of investigating SHC1 binding partners.

RESPONSE: Thanks for your comments and suggestions! According to your comments, we have

reworded the description of the SHCBP1 screening process more logically in the main text. We also performed additional experiments to confirm our screening results. The mRNA expression of the preliminary identified Shc1-binding proteins in tumors from patients with HER2 positive gastric cancer were detected using RT-PCR. We found that Shc1-binding proteins SHCBP1 and RASAL2 were really up-regulated in HER2 positive gastric cancer. The new result was added in the Fig. 1e and the reworded description as follow:

“Using liquid chromatography-tandem mass spectrometry (LC-MS/MS) analysis, we identified 32 Shc1-binding partners including EGFR, HER2, HER3, JUP and SHCBP1 (Fig. 1a). To screen out if any of the binding proteins are potential HER2 downstream regulators, we carried out a gene expression correlation analysis between the identified Shc1 interactors and HER2 expression in 659 gastric cancer specimens obtained from the Gene Expression Omnibus (GEO) database, and 24 HER2-correlated Shc1-binding proteins with spearman coefficient ≥ 0.3 were screened out (Fig. 1b). After that, we determined if any of the identified binding proteins were potential upregulated-oncogenes involved in gastric tumorigenesis. A gene expression profile of gastric cancerous and adjacent normal samples from 16 patients were performed using mRNA microarray. We screened out five overlapping Shc1-binding proteins (JUP, EPHA2, RASAL2, LYN and SHCBP1) which were positively correlated with HER2 expression and were upregulated in gastric cancer (Fig. 1c, d). Finally, to confirm our screening results, the mRNA expression of the identified Shc1-binding proteins in HER2 positive gastric cancer patients were detected using real-time PCR (RT-PCR). We found that Shc1-binding proteins SHCBP1 and RASAL2 were really up-regulated in HER2 positive gastric cancer (Fig. 1e). Of the two binding proteins, we focused on SHCBP1 as it was previously reported interacting with Shc1 prior to EGF stimulation and the role of SHCBP1 in HER2-mediated signal activation was elusive²¹”.

2. Line 96: the term amplified is being used incorrectly.

RESPONSE: Sorry for our mistake. The confusing words “highly amplified” have been corrected as “upregulated”.

3. Gene expression in Fig. 1c is not same as interactions between proteins- not convincing. Of all data

shown, only 1D is somewhat convincing. How other components of HER2 signaling complex correlate with HER2 mRNA. Clarity is needed whether the authors are showing proteins or mRNA vs. copy number.

RESPONSE: Thanks for your suggestion! We have performed additional experiments and rewritten the description of the SHCBP1 screening process according your comments 1 and 3. Please review our response of comments 1.

4. There is no explanation as to why EGF was used: there is no change in IP of SHCBP1 with or without EGF. Plus, EGF does not bind to HER2.

RESPONSE: Thanks a lot! We agree that EGF can bind with HER1 instead of HER2, but HER2 always dimerization with HER1 or other HER members to activate downstream pathways. EGF can bind to HER1 for inducing the heterodimer and phosphorylation of HER1 and HER2 to cause intracellular physiological reaction. EGF was always used to investigate the HER2 activated downstream signaling by others (Triulzi et al, 2018; Jin et al, 2013). Our CO-IP and FRET results between Shc1 and SHCBP1 showed that EGF treatment has a little effect on Shc1-SHCBP1 binding after 5 min while significantly dissociate the Shc1-SHCBP1 interaction after 30 min (Fig. 1f, g). Thus, EGF was used to investigate the function of SHCBP1 in HER2 activated downstream signaling in our study.

References:

Triulzi T, Forte L, Regondi V, et al. HER2 signaling regulates the tumor immune microenvironment and trastuzumab efficacy. *Oncoimmunology*. 2018;8(1): e1512942.

Jin L, Wessely O, Marcusson EG, Ivan C, Calin GA, Alahari SK. Prooncogenic factors miR-23b and miR-27b are regulated by Her2/Neu, EGF, and TNF- α in breast cancer. *Cancer Res*. 2013;73(9):2884-2896.

5. It is unclear if the interaction of SHC1 and its BP1 is a new knowledge obtained in this paper, no citation is provided.

Response: Thanks for your comments! The interaction of Shc1 and SHCBP1 was reported by others and the relevant references were cited in the manuscript (Reference 21).

6. Line 122: how unique is the role of SHCBP1 as a HER2 effector needs to be spelled out. It seems a gross overstatement.

RESPONSE: Thanks a lot! These confusing descriptions have been omitted in the manuscript.

7. Fig. 2: the term “paracancer” is confusing and should be replaced with “adjacent normal” if this is what the authors implied.

RESPONSE: Thanks a lot! The term “paracancer” has been corrected with “adjacent normal” in the figures and the main text. (Fig. 2a, b).

8. Up until Fig. 4, the evidence is guiding to a conclusion that SHCBP1 is an important modulator of HER2 function, thus regulating sensitivity to trastuzumab. However, starting from Fig4, the story completely abandons the idea of HER2 function and goes on to describe SHCBP1 nuclear importation. Proper signaling studies and evaluation of HER2 trafficking and stability in the presence or absence of SHCBP1 are needed to fully exclude the effects on the trastuzumab main target.

RESPONSE: Thanks for your comments! As the schematic diagram shown in Fig. 8, our findings demonstrated that SHCBP1 binds with Shc1 before EGF-induced HER2 activation. However, SHCBP1 is detached from Shc1 following HER2 activation. Then the Shc1 is recruited to bind with HER2 for MAPK or PI3K pathways transduction while SHCBP1 translocates into the nucleus for cell mitosis. That means SHCBP1 acted as a downstream factor of HER2 but has no direct interactions with HER2. SHCBP1, Shc1 and HER2 cannot form a complex at the same time. We also performed additional experiments and confirmed that EGF induced Shc1 membrane colocalization with HER2. SHCBP1 did not colocalize with HER2 but locate into the nucleus (Supplementary Fig 1c). Since Fig. 4, our results indicated that EGF induced the SHCBP1 nuclear translocation and SHCBP1-dependent phosphorylation of MISP for cell mitosis, and blocking HER2 activation using trastuzumab prevented the nuclear translocation of SHCBP1 and phosphorylation of MISP. Inhibiting SHCBP1 nuclear translocation and MISP phosphorylation significantly sensitize gastric cancer to trastuzumab. These findings supported our conclusion that HER2 activation induced the SHCBP1 detached from Shc1 and translocated into the nucleus, and then interacted with PLK1 to promote MISP phosphorylation.

We agree it is possible that the downstream SHCBP1 may have feedback regulation on upstream HER2 activation. We have performed additional experiments to demonstrate the effects of SHCBP1 depletion/overexpression on HER2 phosphorylation but no positive results were obtained (**Fig. 2 for reviewers**).

Fig. 2 for reviewers. Immunoblotting analysis of HER2 and phosphorylated HER2 (*p*-HER2) in SHCBP1 knockdown (shSHCBP1) and SHCBP1 overexpressed (SHCBP1^{+/+}) SNU-216 cells

9. The interpretation of the S273A mutation is not as straightforward as the authors imply: the abrogation of this phosphosite could easily affect its interaction with SHC1/HER2, cause protein misfolding, and not only nuclear translocation. Detailed biochemistry of this mutant is needed to draw this conclusion. In contrast, a more plausible theory could be in place such as stabilization of HER2 or its more robust recycling to the plasma membrane due to its interaction with SHC/SHCBP1. In fact, FRET data (fig.3) shows predominantly membrane colocalization of HER and SHCBP1, nothing is present to appreciable levels in the nucleus.

RESPONSE: Thanks for your suggestions! Our FRET and immunofluorescence colocalization showed that EGF induced Shc1 dissociating from SHCBP1 (Fig. 1g) and Shc1 binding to HER2 (Fig. 1h). We found predominantly membrane colocalization of HER2 and Shc1, not SHCBP1. (Fig. 1h and Supplementary Fig 1c).

To exclude that S273A mutation may cause protein misfolding, we have performed additional experiments and have found that S273A mutation has no effects on SHCBP1-Shc1 interactions and EGF induced SHCBP1-Shc1 dissociation, but blocks the SHCBP1 nuclear translocation (Supplementary Fig. 4d and Fig. 4j). Furthermore, we mutated the Ser273 of SHCBP1 to Arg273, which were more similar biochemically. Our data showed that S273D has no effects on SHCBP1-Shc interactions, but significantly blocked the nuclear translocation of SHCBP1 (Supplementary Fig. 4d, e).

These findings confirmed our conclusion that phosphorylation at the S273 site was indispensable for SHCBP1 nuclear localization. We have added the new findings in Supplementary Fig. 4d, e and the results description were added in the **Results** of the manuscript.

The revised description as follow:

“Immunofluorescence and immunoblotting analyses indicated that the SHCBP1^{S273A} mutant effectively blocked SHCBP1 nuclear translocation but had no effect on EGF induced SHCBP1-Shc1 dissociation (Fig. 4j, Supplementary Fig. 4d). Moreover, we also mutated the Ser273 of SHCBP1 to Arg273, which confirmed that SHCBP1^{S273D} mutant significantly inhibited SHCBP1 nuclear translocation (Supplementary Fig. 4d, e). These results suggested that phosphorylation at the S273 site was indispensable for SHCBP1 nuclear localization.”

10. Functional domains of SHCBP1 are not characterized. No evidence for nuclear localization signal is shown.

RESPONSE: Thanks for your suggestion! We predicted the nuclear localization signal (NLS) using different prediction software and one of them predicted that SHCBP1 623aa-636aa (TQKGQIKKKRLSEL) may be the NLS of SHCBP1. Then, we validated the predicted NLS using protein deletion mutations and nucleoprotein western blotting. However, our data showed that the NLS deletion mutations did not inhibited the nuclear translocation of SHCBP1, suggesting SHCBP1 623aa-636aa is not essential for the SHCBP1 nuclear localization (**Fig. 3 for reviewers**). Based on these results, we speculated that a site modification may serve for SHCBP1 nuclear translocation. Using LC-MS/MS, mutagenesis analysis and nucleoprotein western blotting, we identified that phosphorylation at the S273 site was responsible for SHCBP1 nuclear localization.

Fig. 3 for reviewers. Immunoblotting analysis of SHCBP1 (WT) and SHCBP1 NLS deletion

mutations mutations (1-623aa) in nuclear (Nuc) and cytosol extracts (Cyto) of EGF-treated SNU-216 cells.

11. The abrogation of nuclear fraction of SHCBP1 in cells treated with trastuzumab (Fig. 4E) further argues that BP1 is intimately attached to HER2 signaling complex and is removed from the nucleus by HER2 crosslinking. In these experiments, again, the stimulation of cells with EGF does not provide mechanistic explanation as to why it would change BP1 redistribution between cyto and nuc compartments.

RESPONSE: Thanks a lot! As the schematic diagram shown in Fig. 8. The Shc1 is complex with SHCBP1 in the cytoplasm prior to HER2 activation. Following EGF treatment, Shc1-SHCBP1 complex is disrupted and HER2 recruit Shc1 to a complex in endomembrane. SHCBP1 does not directly interact with HER2 throughout the signaling activation. EGF induced HER2 activation leads to Shc1 leaving from SHCBP1 to bind with HER2 and SHCBP1 is released from Shc1 and then translocated from cytoplasm into the nucleus. Trastuzumab treatment blocks EGF-induced HER2 activation and Shc1 cannot dissociate from SHCBP1. Shc1-SHCBP1 complex in the cytoplasm inhibited the nuclear translocation of SHCBP1.

12. The biochemistry of nuclear interactors with SHCBP1 is interesting although not convincing the mechanism of TF3 is necessarily related to its action on nuclear proteins. Again, in line with the comments on mechanism above, trastuzumab and TF3 combo produces remarkable HER2 depletion (Fig.6), which is a likely mechanism for the synergy. Testing of TF3 in disrupting SHC1/SHCBP1/HER2 complexes will clarify and support this more interesting and signaling-specific mechanism. Additional detailed signaling and HER2 trafficking characterization is needed for the TF3 mechanism.

RESPONSE: Thanks for your comments! As mentioned above, Shc1 can interact with SHCBP1 prior to EGF stimuli or bind with HER2 after EGF treatment. SHCBP1-Shc1-HER2 cannot be a complex at the same time. The overexpressed SHCBP1 can hyperactivate downstream SHCBP1-PLK1-MISP signaling and impairs trastuzumab sensitivity. Thus, we identified TFBG (TF3) as a downstream inhibitor of SHCBP1-PLK1 complex to block SHCBP1-PLK1 induced cell division. TFBG and

trastuzumab can synergistically inhibit tumor growth by inhibiting upstream of HER2 with trastuzumab and downstream of SHCBP1-PLK1 with TFBG.

According to your suggestion, to validate whether there are feedback effects of TFBG on upstream of Shc1-SHCBP1 interaction, we performed additional experiments. Our data showed that TFBG can inhibit the EGF-induced Shc1-SHCBP1 dissociation. These results suggested that inhibiting the downstream SHCBP1-PLK1 using TFBG induced a feedback inhibition on upstream Shc1-SHCBP1. These new findings have been added in the Supplementary Fig. 8h and the description was added in the main text:

“To further investigate whether TFBG blocking SHCBP1-PLK1 interaction has feedback effects on the upstream Shc1-SHCBP1 bindings, we detected the Shc1-SHCBP1 interactions after 5 μ M and 10 μ M TFBG treatment. We found that TFBG partly suppressed EGF-induced Shc1 and SHCBP1 dissociation, which suggested that the feedback inhibition on Shc-SHCBP1 of TFBG is one of the reasons why TFBG sensitize gastric cancer to trastuzumab (Supplementary Fig. 9h).”

Reviewers' Comments:

Reviewer #1:

Remarks to the Author:

There is one mistake in newly added text in the revised version. On page 9, line 225, it should read: "Asp273" instead of "Arg273".

Other than that, the authors have addressed all my comments satisfactorily and I recommend publication.

Reviewer #2:

None

Reviewer #3:

Remarks to the Author:

The authors thoroughly revised the manuscript to address the deficiencies identified by this Reviewer and other Reviewers. The concept that SHCBP1 binds to the mitotic spindle following HER2 heterodimerization with EGFR is based on a solid body of biochemical evidence. The paper provides an important finding that SHCBP1 is a mitotic effector of HER2 via a direct physical contact with the mitotic kinase PLK1.

One remaining weak point in the paper is the introduction of theaflavin digallate as an inhibitor of PLK1-SHCBP1 interaction. Plasmon resonance and FRET produce the results suggest for the interaction, but the evidence that theaflavin binds to the pocket on the surface of PLK1, AND prevents SHCBP1 loading onto PLK1 is weak. More definitive experiments such as e.g. disruption of the two proteins pulldown, is needed. TF3 has been imparted with numerous effects in the published literature. It is unclear whether theaflavin-digallate binding to PLK1 affects PLK1 function directly, and how the interaction of PLK1 and SHCBP1 is important in its cytotoxic effect. Previous papers (doi: 10.1002/mc.20174) show its direct effect on EGFR downregulation, which is in part may be responsible for its augmentation of trastuzumab activity in HER2-amplified gastric cancer cell lines.

Responses to Referees Letter
(NCOMMS-20-38715B)

.....

Responses to reviewer comments:

Reviewer #1:

There is one mistake in newly added text in the revised version. On page 9, line 225, it should read: "Asp273" instead of "Arg273".

Other than that, the authors have addressed all my comments satisfactorily and I recommend publication.

RESPONSE: Sorry for our mistake and it has been corrected.

Reviewer #3:

The authors thoroughly revised the manuscript to address the deficiencies identified by this Reviewer and other Reviewers. The concept that SHCBP1 binds to the mitotic spindle following HER2 heterodimerization with EGFR is based on a solid body of biochemical evidence. The paper provides an important finding that SHCBP1 is a mitotic effector of HER2 via a direct physical contact with the mitotic kinase PLK1.

One remaining weak point in the paper is the introduction of theaflavin digallate as an inhibitor of PLK1-SHCBP1 interaction. Plasmon resonance and FRET produce the results suggest for the interaction, but the evidence that theaflavin binds to the pocket on the surface of PLK1, AND prevents SHCBP1 loading onto PLK1 is weak. More definitive experiments such as e.g. disruption of the two proteins pulldown, is needed. TF3 has been imparted with numerous effects in the published literature. It is unclear whether theaflavin-digallate binding to PLK1 affects PLK1 function directly, and how the

interaction of PLK1 and SHCBP1 is important in its cytotoxic effect. Previous papers (doi: 10.1002/mc.20174) show its direct effect on EGFR downregulation, which is in part may be responsible for its augmentation of trastuzumab activity in HER2-amplified gastric cancer cell lines.

RESPONSE: Thanks for your comments! We have performed GST pull-down experiment according to your suggestion. The results are consistent with our existing CO-IP and FRET data that TFBG is an inhibitor of SHCBP1-PLK1 interaction, demonstrating the disruption effects of TFBG on SHCBP1-PLK1 complex (Supplementary Fig. 8f).

TFBG was screened out based on the pocket of PLK1 which interacts with SHCBP1, not the PLK1 catalytic or substrate-binding pocket. However, it is possible that TFBG may also inhibit the binding of PLK1 with other proteins. We have added statements in this regard in the *Discussion* part. We agree that TFBG may also target to other signal pathways including EGFR, which may partly be responsible for the sensitization of TFBG to trastuzumab. We have added this in the *Discussion* part and the relevant reference was cited (Reference 43).

Reviewers' Comments:

Reviewer #3:

Remarks to the Author:

Pull down experiments show competitive blockade of SHCBP1-PLK1 interaction. No further corrections are needed. Thank you.

Reviewer #4:

Remarks to the Author:

Impressive job! And I have some questions about the details.

1. In the combination therapy, what is the combination index(CI) of the two drugs?
2. In the in vivo tumor inhibition experiment, the data only showed the inhibition effects of two cell lines? How about the effect of patient-derived xenografts mouse model of gastric cancer?
3. S273 of SHCBP1, S394A, S395A and S397A of MISP, K474S, Y485F, H489N and L490A of PLK1, are mutations of these sites existing in gastric specimens?
4. Has the toxicity of TFBG been tested? How to choose the dosages of the two drugs?
5. SHCBP1 can also interact with PTEN, is it considered as a crosstalk between HER2-PLK1-MISP and HER2-AKT?
6. Does the sublocation of SHCBP1 vary in gastric cancer of different pathological types or different stages?

Responses to Referees Letter (NCOMMS-20-38715C)

.....

Responses to reviewer comments:

Reviewer #4:

Impressive job! And I have some questions about the details.

1. In the combination therapy, what is the combination index (CI) of the two drugs?

RESPONSE: Thanks a lot! The combination index (CI) equation is based on the multiple drug-effect equation of Chou-Talalay derived from enzyme kinetic models, which is widely used to determine the synergism or antagonism of combinations with two drugs. CI values < 1 indicates synergism, CI values > 1 indicates antagonism and CI = 1 indicates additivity. Herein, cells were treated with different concentrations of trastuzumab (from 1 nM to 50 μ M) and TFBG (0 μ M, 10 μ M and 20 μ M) and the cell viability was detected, and then the CI values and the fraction affected (Fa) were analyzed using CalcuSyn software (BioSoft). The synergistic effects of trastuzumab and TFBG are indicated as the CI is significantly less than 1 (Supplementary Fig. 9A).

2. In the in vivo tumor inhibition experiment, the data only showed the inhibition effects of two cell lines? How about the effect of patient-derived xenografts mouse model of gastric cancer?

RESPONSE: Thanks for your suggestions! Validation of the tumor inhibition of trastuzumab and TFBG combination using patient-derived xenografts mouse model is valuable. We are trying to establish a HER2 positive PDX model and the tumor inhibition experiment will be performed in the future. We have added a statement in this regard in the *Discussion* part.

3. S273 of SHCBP1, S394A, S395A and S397A of MISP, K474S, Y485F, H489N and L490A

of PLK1, are mutations of these sites existing in gastric specimens?

RESPONSE: Thanks for your comments! We have searched the mutations of SHCBP1, MISP and PLK1 in gastric cancer using an online database [International Cancer Genome Consortium (ICGC), <https://dcc.icgc.org/>] and no mutations of above amino acid sites were found in gastric cancer.

4. Has the toxicity of TFBG been tested? How to choose the dosages of the two drugs?

RESPONSE: Thanks for your comments! Trastuzumab was used for the treatment of HER2 positive cancers widely and we chose the dosage as 10 mg/kg according to the reference (Wainberg. et al. 2010). 50 mg/kg TFBG was chosen as a safe dose which was used for other animal experiments by others (Teng. et al. 2019). After the treatments of trastuzumab and TFBG combination, we have not found significant weight loss of the animals (Supplementary Fig 9c). However, we agree that systematic toxicological testing is necessary for drugs reaching the clinic and have added a statement in this regard in the *Discussion* part.

References:

Wainberg, Z.A. et al. a dual EGFR and HER2 kinase inhibitor, selectively inhibits HER2-amplified human gastric cancer cells and is synergistic with trastuzumab in vitro and in vivo. *Clin Cancer Res* **16**, 1509-1519 (2010).

Teng, Z. et al. Theaflavin-3,3'-digallate increases the antibacterial activity of β -lactam antibiotics by inhibiting metallo- β -lactamase activity. *J Cell Mol Med* **23**, 6955-6964 (2019).

5. SHCBP1 can also interact with PTEN, is it considered as a crosstalk between HER2-PLK1-MISP and HER2-AKT?

RESPONSE: Thanks for your comments! It was reported that silencing of SHCBP1 expression led to an increase of PTEN and cell apoptosis of lung cancer. Thus, it is possible that SHCBP1 synergistically activates PI3K/AKT signaling by inhibition of PTEN. Revealing the mechanism of the crosstalk between HER2/SHCBP1/PLK1 and PI3K/AKT will be valuable in the future. We have added a statement in the *Discussion* part and the reference was cited (Reference 38).

6. Does the sublocation of SHCBP1 vary in gastric cancer of different pathological types or different stages?

RESPONSE: Thanks for your comments! Our immunohistochemistry result of gastric cancer TMA showed that there is no significant correlation between SHCBP1 expression with tumor

pathological types ($p = 0.901$) but significant correlated with tumor invasion ($p = 0.04$, Supplementary Fig 2.c). These have been described in the *Result* of the manuscripts. We agree that the detection of the sublocation of SHCBP1, such as the nuclear localization, is also meaningful and it will be detected in the future.